# CNN Interpretability with Multivector Tucker Saliency Maps for Self-Supervised Models

**Aymene Mohammed Bouayed**                                          *aymene.bouayed@ens.fr*
*DIÉNS, ÉNS, CNRS, PSL University, Paris, France*
*Be-Ys Research, France*

**Samuel Deslauriers-Gauthier**                          *samuel.deslauriers-gauthier@inria.fr*
*Centre Inria d'Université Côte d'Azur, Nice, France*

**Adrian Iaccovelli**                                              *adrian.iaccovelli@be-ys.com*
*Be-Ys Research, France*

**David Naccache**                                                *david.naccache@gmail.com*
*DIÉNS, ÉNS, CNRS, PSL University, Paris, France*

Reviewed on OpenReview: *https://openreview.net/forum?id=3605*

## Abstract

Interpreting the decisions of Convolutional Neural Networks (CNNs) is essential for understanding their behavior, yet it remains a significant challenge, particularly for self-supervised models. Most existing methods for generating saliency maps rely on reference labels, restricting their use to supervised tasks. EigenCAM is the only notable label-independent alternative, leveraging Singular Value Decomposition to generate saliency maps applicable across CNN models, but it does not fully exploit the tensorial structure of feature maps. In this work, we introduce the Tucker Saliency Map (TSM) method, which applies Tucker tensor decomposition to better capture the inherent structure of feature maps, producing more accurate singular vectors and values. These are used to generate high-fidelity saliency maps, effectively highlighting objects of interest in the input. We further extend EigenCAM and TSM into multivector variants—Multivec-EigenCAM and Multivector Tucker Saliency Maps (MTSM)—which utilize all singular vectors and values, further improving saliency map quality. Quantitative evaluations on supervised classification models demonstrate that TSM, Multivec-EigenCAM, and MTSM achieve competitive performance with label-dependent methods. Moreover, TSM enhances interpretability by approximately 50% over EigenCAM for both supervised and self-supervised models. Multivec-EigenCAM and MTSM further advance state-of-the-art interpretability performance on self-supervised models, with MTSM achieving the best results.

## 1 Introduction

Convolutional Neural Networks (CNNs) demonstrate exceptional performance across various computer vision applications (Redmon et al., 2016; Krizhevsky et al., 2017; Liu et al., 2021). These networks can be trained in a supervised manner to tackle classification or regression tasks (He et al., 2016) or in a self-supervised manner for image segmentation (Walsh et al., 2022; Kirillov et al., 2023), face recognition (Boutros et al., 2022), and learning image embeddings (Bardes et al., 2022). However, their inherent black-box nature poses significant challenges, particularly in medicine (Cruciani et al., 2021; Bouayed et al., 2022) and biometrics (Kim & Cho, 2021), where interpretability is crucial for trust and adoption.

Numerous works have attempted to address the challenge of interpreting CNN decisions, proposing diverse methods that can be broadly categorized into feature attribution methods and Class Activation Map (CAM)

methods. Feature attribution methods involve either estimating a locally interpretable model (Ribeiro et al., 2016) or propagating information from the network's output to its input using gradients (Springenberg et al., 2015), contribution scores (Bach et al., 2015; Shrikumar et al., 2017), or Shapley values (Lundberg & Lee, 2017). However, these methods often come with significant computational overhead, even when using approximations (Aas et al., 2021). In contrast, CAM methods (Zhou et al., 2016; Chattopadhay et al., 2018) perform a weighted sum of the feature map tensors, a multidimensional array which preserves spatial relationships, and avoid backward propagation, yet they require a reference class to infer the weights (Chattopadhay et al., 2018; Wang et al., 2020; Desai & Ramaswamy, 2020; Schulz et al., 2020; Zhang et al., 2024).

Despite these advancements, a major limitation of current interpretability methods is their predominant focus on supervised classification models. Most existing techniques rely heavily on reference labels to generate saliency maps, rendering them label-dependent and restricting their applicability to supervised tasks. The only exception is EigenCAM (Muhammad & Yeasin, 2020), which offers a label-independent approach and can be applied to both supervised and self-supervised models. However, EigenCAM has notable drawbacks: its use of matricization[1] disrupts the spatial relationships within the feature maps, particularly in the height and width dimensions. This limitation leads to less accurate estimations of singular vectors and negatively impacts the quality of the resulting saliency maps.

In this work, we aim to propose a general label-independent interpretability method for CNNs that addresses the limitations of existing approaches, particularly EigenCAM. Specifically, we seek to answer the question: "*Can tensor algebra provide a better estimation of the weight vector?*". We answer the question positively through the introduction of the *Tucker Saliency Map* (TSM) method. TSM applies Tucker tensor decomposition (Kolda & Bader, 2009; Cheng et al., 2023) directly to the feature map tensor, preserving spatial relationships and resulting in different singular vectors. This enhancement allows for improved saliency maps that capture more variance in the first singular vector.

Furthermore, we extend both EigenCAM and TSM into multivector variants: *Multivec-EigenCAM* and *Multivector Tucker Saliency Maps* (MTSM), which leverage all singular vectors and values to create richer and more accurate saliency maps than their respective base methods.

We evaluate our proposed methods in both label-dependent and label-independent settings. In the label-dependent context, TSM, Multivec-EigenCAM, and MTSM demonstrate competitive performance against six state-of-the-art label-dependent CAM methods while showing substantial improvements over EigenCAM. For the label-independent setting, we introduce a comprehensive evaluation framework tailored for self-supervised models, addressing a significant gap in the literature. This framework assesses the impact of saliency maps on downstream classification performance, the reproducibility of generated embeddings, and their similarity to ground truth segmentation maps. Our results consistently show that TSM outperforms EigenCAM, establishing a new state-of-the-art in self-supervised CNN interpretability. Multivec-EigenCAM and MTSM further enhance these results, particularly with MTSM leading in performance.

We outline our contributions as follows :

- We introduce Tucker Saliency Map (TSM) for label-independent CNN interpretability, leveraging Tucker tensor decomposition to enhance saliency map quality.
- We develop Multivec-EigenCAM and Multivector Tucker Saliency Maps (MTSM) to utilize all singular vectors for richer saliency maps.
- We introduce a new framework for assessing saliency maps in self-supervised models, focusing on classification performance, embedding reproducibility and alignment with segmentation masks.
- We evaluate the proposed methods in both label-dependent and label-independent settings, showing competitive performance against state-of-the-art methods. Also, we establish TSM as a new state-of-the-art in self-supervised interpretability, with MTSM furthering the state-of-the-art performance.

The remained of this paper is organized as follows: in Section 2 we introduce essential background definitions and notations. Section 3 reviews related work addressing various interpretability methods for CNN models.

---

[1]Matricization is the operation of transforming a tensor into a matrix (Kolda & Bader, 2009).

The mechanics of our proposed methods are detailed in Sections 4 and 5. We rigorously test and verify the efficacy of our methods across a diverse set of supervised and self-supervised CNN models in Section 6. Finally, we conclude and outline potential avenues for future exploration in Section 7.

## 2 Background

### 2.1 Class Activation Maps

Class Activation Maps (CAMs) (Zhou et al., 2016) are local interpretability methods for CNN based models. They help visualize the regions of the input image that significantly contribute to the CNN's prediction by analyzing the intermediate results. Formally, CAMs are defined as the sum of the feature maps resulting from the last convolutional layer of a CNN $\mathcal{F} \in \mathbb{R}^{C,H,W}$, weighted by a vector $w \in \mathbb{R}^C$ as follows:

$$\text{CAM} = \sum_{i=1}^{C} w_i \mathcal{F}_i. \tag{1}$$

As observed from Equation 1, the weight vector $w$ is the only unknown to determine[2]. Various works propose different methods to infer $w$ (Chattopadhay et al., 2018; Wang et al., 2020; Muhammad & Yeasin, 2020; Zhang et al., 2024). In Section 3.2, we provide an overview of the different CAM based methods with a focus on the EigenCAM method (Muhammad & Yeasin, 2020) which utilizes the SVD (Trefethen & Bau, 1997).

### 2.2 Matrix Decomposition and SVD

Matrix decomposition involves factorizing a matrix $\mathcal{M} \in \mathbb{R}^{m,n}$ into a product of $k$ matrices to simplify computations and reveal underlying structures (Trefethen & Bau, 1997). Singular Value Decomposition (SVD) (Trefethen & Bau, 1997) is a prominent matrix decomposition method, which expresses a matrix $\mathcal{M}$ as $\mathcal{M} = U\Sigma V^{\intercal}$. Matrices $U \in \mathbb{R}^{m,m}$ and $V \in \mathbb{R}^{n,n}$ consist of *singular vectors* and form a basis for the rows and columns of $\mathcal{M}$, respectively. The matrix $\Sigma \in \mathbb{R}^{m,n}$ is a diagonal matrix containing *singular values* on the diagonal, representing the information and variance encoded per direction of the new basis (Trefethen & Bau, 1997). However, the SVD is exclusively applicable to matrices therefore can not be applicable to higher dimensional data in the form of tensors without the need to undergo a matricization operation. This ultimately results in a loss of relationship information between the elements of flattened modes. Consequently, tensor algebra introduces a multitude of decompositions which operate directly in the tensor space hence solving the relationship disruption problem.

### 2.3 Tensors and Tucker Decomposition

A tensor $\mathcal{T} \in \mathbb{R}^{n_1,n_2,\dots,n_k}$ an algebraic object that can be thought of as a multi-dimensional array that generalizes matrices to higher dimensions, allowing for more complex data representations. Consequently, a multitude of operations on matrices can be extended to tensors. One such extension is *tensor decomposition* (Kolda & Bader, 2009). Among the most well known tensor decompositions we find the *CANDECOMP/PARAFAC* (CP) decomposition (Kiers, 2000; Bove, 2010) and the *Tucker tensor decomposition* (Sheehan & Saad, 2007; Cheng et al., 2023). As noticed by Kolda & Bader (2009), CP decomposes a tensor as a sum of rank-one tensors with no additional constraints, whereas the Tucker tensor decomposition is a higher-order form of principal component analysis. Hence, the Tucker decomposition provides sorted singular vectors according to singular values as opposed to the CP decomposition. This property is of importance to the modeling of our proposed TSM methods and motivates our choice of the Tucker decomposition over the CP decomposition or any other decomposition (See Section 4).

Focusing on the Tucker tensor decomposition, it decomposes a tensor $\mathcal{T}$ as :

$$\mathcal{T} = \mathcal{C} \times_1 A^{(1)} \times_2 A^{(2)} \times_3 \cdots \times_k A^{(k)}. \tag{2}$$

---

[2]Additional operations can be applied to the weighted sum depending on the CAM method. Among such operations is the ReLU function and/or the min-max normalization.

Here, $\mathcal{C} \in \mathbb{R}^{n_1, n_2, \ldots, n_k}$ is the core tensor containing singular values calculated by taking the Frobenius norm along each mode $i$. The symbol $\times_i$ denotes matrix multiplication along the $i^{\text{th}}$ mode, and $A^{(1)}, \ldots, A^{(k)}$ are unitary matrices representing the newly calculated basis, akin to the matrices $U$ and $V$ in SVD.

The estimation of the values of the core tensor $\mathcal{C}$ and the basis $\{A^{(i)}\}_{i=1}^k$ can be achieved through different algorithms, notably, the Higher Order Orthogonal Iteration (HOOI) algorithm (Sheehan & Saad, 2007) or the Higher-Order SVD (HOSVD) (De Lathauwer et al., 2000) algorithm. In this work, we harness the HOOI algorithm, as it is known to be more accurate than HOSVD for Tucker Decomposition. HOOI uses HOSVD as an initialization step increasing the chances of having a unique solution which is then optimized in order to minimize the the Frobenius form between the tensor $\mathcal{T}$ and its decomposition (Kolda & Bader, 2009).

## 3 Related work

### 3.1 Feature attribution methods

Feature attribution methods are interpretability techniques which assign a weight to each feature of the input, such as a pixel in an image (Lundberg & Lee, 2017). One such feature attribution method is the *Guided Backpropagation* method (Springenberg et al., 2015), which propagates positive gradients of the classification loss function to the input. However, to address the gradient saturation and the stochastic nature of gradients, methods like *Layer-wise Relevance Propagation (LRP)*(Bach et al., 2015) and DeepLFT (Shrikumar et al., 2017) were proposed. These methods calculate a relevance score for the output of each neuron $j$ as a weighted sum of all the neurons it is connected to. This operation is then back-propagated till the input, resulting in a heatmap showcasing the importance of each region to the network's output (Montavon et al., 2019). However, Lundberg & Lee (2017) notices that the latter methods require domain-specific knowledge, which is dataset-specific, and therefore introduces *Deep SHapley Additive exPlanations (Deep SHAP)* (Lundberg & Lee, 2017). Deep SHAP uses Shapley values (Ichiishi, 1983) to calculate the contribution of each feature in the network's input. Despite its strong theoretical backing, Deep SHAP's computational complexity poses challenges for large datasets, making it less practical for real-time applications, even with approximation methods (Aas et al., 2021). Moreover, all the stated methods highlights certain pixels which does not produce spatially human-interpretable coherent maps. LIME (Ribeiro et al., 2016) solves the latter problem and highlights contiguous regions of the input. This is achieved by perturbing input data and create a locally interpretable model around the predicted classification, relying on the model's output class.

All feature attribution methods are inherently label-dependent, as they calculate saliency maps based on a specified target class in different ways. The reliance of feature attribution methods on label information prevents their application to self-supervised models, where no labels are available. In contrast, our proposed methods, TSM and MTSM, operate in a label-independent manner and produce visually interpretable saliency maps by highlighting larger, contiguous regions of the input. Consequently, in our experiments we exclusively evaluate the performance of feature attribution methods on classification models.

### 3.2 Class activation map methods

Building on feature attribution techniques, Class Activation Maps (CAMs) (Zhou et al., 2016) offer a visual interpretation of model predictions, particularly in computer vision tasks. They have also found application in different domains, such as anomaly detection (Kimura et al., 2020), disease identification and localization (Khan et al., 2019; Bouayed et al., 2022). CAM methods can be broadly categorized into *label-dependent* and *label-independent* methods. Below is a brief description of each family of CAMs.

#### 3.2.1 Label-dependent class activation maps

Label-dependent class activation maps rely on the label information to generate saliency maps. The label information is combined with the output of the network to estimate a loss and backpropagate the loss information to infer the weight vector $w$. This information can be in the form of gradients as in GradCAM (Selvaraju et al., 2017) and XGradCAM (Fu et al., 2020), or the impact of each feature map on the classification of the input image in the class specified label as in ScoreCAM (Wang et al., 2020) and

AblationCAM (Desai & Ramaswamy, 2020). Other methods perform an optimization procedure on $w$ to minimize the model's training loss w.r.t. the specified label namely Information Bottleneck Attribution (Schulz et al., 2020) and Opti-CAM (Zhang et al., 2024).

These methods primarily target classification models, providing interpretations in the form of a saliency map for a given reference label. Additionally, the most widely used label depend methods are gradient methods such as GradCAM (Selvaraju et al., 2017). However, since gradients indicate the direction of the steepest decent and the loss function is non-linear, the communicated information by the gradient changes rapidly. This can be seen as noise which prevents the accurate estimation of the weight vector and the saliency map.

Recent work (Ramprasaath R. Selvaraju, 2021; Shu et al., 2023) use the gradient of the contrastive loss to calculate the GradCAM based saliency map of a self-supervised model. However, the accurate estimation of the contrastive loss heavily depends on the number of negative samples. Therefore, it is required to have a large batch size as in (Chen et al., 2020a) or a queue storing previous data points seen during the training as in (Chen et al., 2020b; Caron et al., 2020). Moreover, data augmentation techniques are essential to the generation of negative samples and some data augmentation transformations allow for more informative contrastive loss than others (See Figure 5 in (Chen et al., 2020a)). However, in the absence of a large batch size (in inference for instance), loss estimation will degrade and the gradient of the loss would be even noisier. Consequently, using this gradient in a GradCAM-like method would result in poor-quality saliency maps.

In view of these constraints, using gradient based CAM methods for the interpretability of self-supervised CNN models would results in saliency maps which depend on the careful tunning of multiple factors. As a result, in our experiments on label-dependent methods we only consider classification models.

### 3.2.2 Label-independent class activation maps

Label-independent class activation maps rely on the decomposition of the feature map tensor to infer the weight vector $w$. To the best of our knowledge, EigenCAM (Muhammad & Yeasin, 2020) is the only method in this family. EigenCAM flattenes the feature map tensor into a matrix, and after applying the SVD (Trefethen & Bau, 1997), the first right singular vector is used as the weight vector. Despite EigenCAM's good performance, the matricization step disrupts spatial relationships between dimensions, captures a limited amount of variance in the first singular vector and fails to utilize the full potential of tensor algebra, leading to a less accurate weight vector and a lower quality saliency map. To address these shortcomings, in the following Section 4 we propose the TSM method, which leverages Tucker tensor decomposition for more accurate weight vector estimation and improved saliency maps. Moreover, since both EigenCAM and TSM rely only on one singular vector, in Section 5 we propose an extension of both methods to harness all the singular vectors and singular values.

## 4 Tucker Saliency Maps

The Tucker Saliency Map (*TSM*) method is a label-independent local interpretability approach. It involves generating saliency maps in the form of heat maps highlighting significant regions in the input images of a CNN related to the task at hand. To accomplish this, firstly the feature map tensor outputted by a convolutional layer is retrieved. Then, a weighted sum of this tensor is performed along the channels mode with the weight vector weighting each feature map estimated through the Tucker tensor decomposition. Consequently, it can be noticed that TSM is applicable to both supervised and self-supervised CNN models as it does not require information regarding the loss but only requires access to the feature map tensor. Figure 1 provides an overview of the proposed method, visually summarizing its key components and process.

In detail, given a feature map tensor $\mathcal{F} \in \mathbb{R}^{C,H,W}$ retreived as the output of a convolutional layer, we firstly perform its Tucker tensor decomposition :

$$\mathcal{F} = \mathcal{C} \times_1 A^{(1)} \times_2 A^{(2)} \times_3 A^{(3)} \tag{3}$$

With $\mathcal{C}$ being the core tensor, $A^{(1)} \in \mathbb{R}^{C,C}$, $A^{(2)} \in \mathbb{R}^{H,H}$, and $A^{(3)} \in \mathbb{R}^{W,W}$ are the matrices containing the orthonormal singular vectors for each dimension. The choice of Tucker tensor decomposition among all tensor

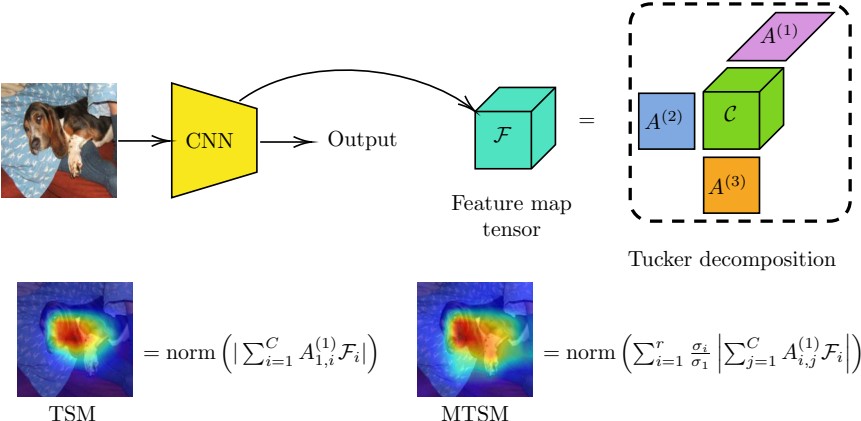

Figure 1: Proposed TSM and MTSM methods. Firstly, the feature map tensor $\mathcal{F}$ is extracted from the output of a convolutional layer (preferably the last convolutional layer, as it has the widest field of view). Then, the Tucker tensor decomposition (Sheehan & Saad, 2007) is performed on $\mathcal{F}$. This decomposition results in a core tensor $\mathcal{C}$ and three matrices containing orthonormal singular vectors for each dimension, notably $A^{(1)}$, $A^{(2)}$, and $A^{(3)}$. The matrix $A^{(1)}$ and the singular values are appropriately used to calculate the TSM (See Equation 4) and the MTSM (See Equation 6). Finally, the absolute value function is applied per element to the resulting matrix and the values are renormalized to be in the range $[0; 1]$.

decomposition methods is strategic, as it accurately identifies the direction of highest variance, a feature crucial for weighting feature maps effectively (Kolda & Bader, 2009).

Secondly, inspired by the work of Muhammad & Yeasin (2020), we consider the first singular vector i.e. the one associated to the largest singular value, $A_1^{(1)}$ of the matrix $A^{(1)}$ as the weight vector $w \in \mathbb{R}^C$. This can be seen as projecting the feature map tensor on the direction with the highest variance. As a result, the obtained saliency maps highlighting more of the regions important to the model's output hence more accurate local interpretability. Formally, we calculate the saliency map as follows :

$$\mathcal{C}, A^{(1)}, A^{(2)}, A^{(3)} = \text{Tucker}(\mathcal{F}) \quad , \quad w = A_1^{(1)} \tag{4}$$

$$TSM = \text{norm}\left(\left|\sum_{i=1}^{C} w_i \cdot \mathcal{F}_i\right|\right).$$

The application of min-max normalization $\text{norm}(\cdot)$ ensures that the saliency map's values are bounded within the $[0, 1]$ range. The absolute value function $|\cdot|$ is employed instead of ReLU to focus on the activation magnitude, disregarding the sign, considering the possibility of negative value multiplication in subsequent layers of the feature map $\mathcal{F}$.

The main difference between TSM and EigenCAM resides in the decomposition methods. EigenCAM performs an SVD decomposition after a matricization operation of the feature map tensor $\boldsymbol{mat(\cdot)}$ :

$$U, \Sigma, V^\intercal = \text{SVD}(\boldsymbol{mat}(\mathcal{F})) \quad , \quad w = V_1. \tag{5}$$

The matricization operation on $\mathcal{F}$ ultimately mixes the elements of the first and second mode (height and width) so as to obtain a matrix of size $\mathbb{R}^{C, H \times W}$. Such operation would have a negative impact on the relationships between the elements of the tensor. Therefore, performing a decomposition on such matrix results in less accurate estimation of the singular vectors and values for each mode which leads to less representative saliency maps. However, in TSM, we adopt a more general and more suitable neighborhood information preserving decomposition notably the Tucker tensor decomposition. This decomposition allows for a better approximation of the first singular vector capturing the most variance. This can be observed in Figure 2 where the amount of variance encoded by the first singular vectors using the Tucker decomposition

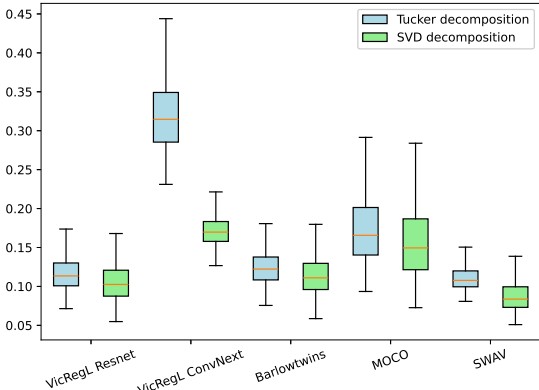

Figure 2: Boxplot comparing the distribution of the first singular values divided by the sum of all singular values per tensor in both SVD decomposition and Tucker decomposition. These singular values are obtained from the decomposition of all the feature map tensors produced by five different self-supervised models on the Pascal VOC (Everingham et al., 2010) validation dataset.

is higher compared to the SVD decomposition on most tested models. Additionally, the use of the absolute value in the calculation of the saliency map puts forward more information hence more informative and accurate saliency maps. Nevertheless, both EigenCAM and TSM utilize only the singular vector associated with the largest singular value. In the following section, we propose a generalization of both methods to utilize all singular values and singular vectors.

## 5 Multivector EigenCAM and Multivector Tucker Saliency Maps

Figure 3 represents the distribution of the first five singular values in both SVD decomposition and Tucker decomposition divided by the sum of all singular values per tensor. From this figure we notice that for both the SVD and Tucker decompositions singular values other than the first one encapsulate a significant amount of variance. Consequently, EigenCAM and TSM only convey a portion of the total information encoded in the feature map tensor. To this end, in this section we introduce Multivector EigenCAM and Multivector Tucker Saliency Maps (*Multivec-EigenCAM* and *MTSM*) interpretability methods. These methods generate a saliency map per singular vector in the SVD or the Tucker tensor decomposition. Then, a weighted sum of the generated saliency maps is performed. The weights of each saliency map are inferred based on the singular values such as, given $\sigma_i$ the singular values associated with the $i$-th singular vector (extracted either from the SVD or the Tucker decomposition), the weight are given by $\frac{\sigma_i}{\sigma_1}$ where $\sigma_1$ is the largest singular value. We provide an overview of the proposed MTSM method in Figure 1.

Formally, given a feature map tensor $\mathcal{F} \in \mathbb{R}^{C,H,W}$, an operator $D(\cdot)$ encapsulating the SVD or Tucker tensor decomposition and returning the ordered singular values $\sigma \in \mathbb{R}^r$ and their corresponding singular vectors for the channels dimension $V \in \mathbb{R}^{r,C}$ :

$$\sigma, V = D(\mathcal{F}) \tag{6}$$

$$\text{Mutivec-*} = \text{norm}\left(\sum_{i=1}^{r} \frac{\sigma_i}{\sigma_1} \left| \sum_{j=1}^{C} V_{i,j}\mathcal{F}_i \right| \right).$$

This formulation of the Mutivec-* can be seen as first generating the EigenCAM or TSM, since the weight of the first saliency map is one, than adding additional information to it which is generated using the rest of the singular vectors. Consequently, the produced saliency maps are richer and more precise as demonstrated quantitatively and qualitatively in Section 6.

The differentiating point between uni-vector methods to its multi-vector ones in performance would depend on the amount of variance encoded by the singular vectors other than the first one. If these vectors do not

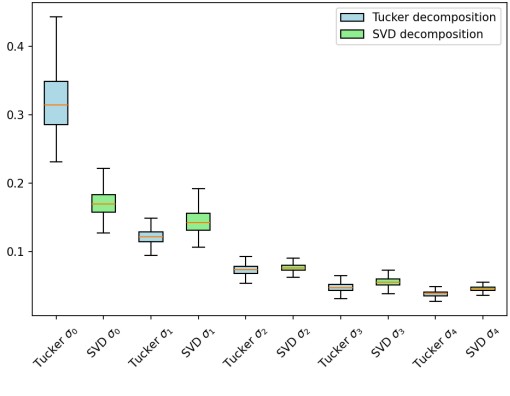

(a) VicRegL ConvNext (Bardes et al., 2022)

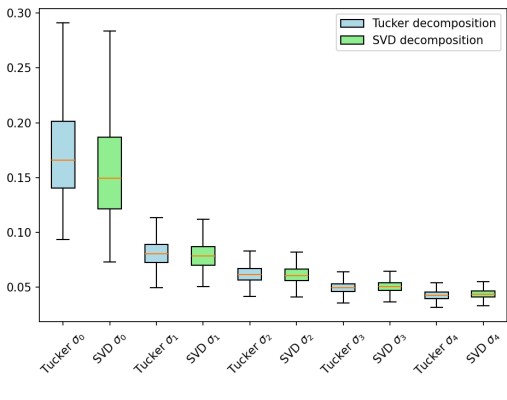

(b) Moco V2 (Chen et al., 2020b)

Figure 3: Boxplot representing the distribution of the first five singular values in both SVD decomposition and Tucker decomposition divided by the sum of all singular values per tensor. These singular values are obtained from the decomposition of all the feature map tensors produced by the Moco V2 (Chen et al., 2020b) and VicRegL ConvNext (Bardes et al., 2022) models on the Pascal VOC validation dataset.

encode a large amount of variance the performance does not improve significantly (See the Tucker tensor decomposition in Figure 3(a) and the quantitative results in Table 2). However, if they encode a large amount of information, harnessing these vectors would result in significant performance improvement (See Figure 3(b)). Consequently, multi-vector saliency map methods in the worst scenario output the same result as the uni-vector methods. Moreover, as SVD based saliency methods do not estimate the most accurate singular vectors, we notice a significant amount of variance encoded by the rest of the singular vectors in Figure 3. Therefore, EigenCAM's performance would greatly improve when opting for the multi-vector formulation. We quantitatively and qualitatively confirm these remarks in the following Section 6.

## 6 Experiments

In this section we start by presenting the metrics we use to evaluate interpretability methods based on saliency maps both on supervised classification models and self-supervised representation learning models. Then, we present, discuss and compare the experimental results of our proposed methods compared to methods in the literature. Moreover, in Appendix A we present Python code for the implementation of our proposed methods and in Appendix B we lay out our experimental setup. Appendices E, D and F put forward additional qualitative results for the different tested supervised and self-supervised CNN models.

### 6.1 Evaluation metrics

### 6.1.1 Supervised learning metrics

To evaluate our proposed methods on supervised models, we opt for the two conventionally used metrics, the Average Drop and Average Increase proposed in the work of Chattopadhay et al. (2018) as they align with human interpretation and favor saliency maps with contiguous regions.

- **Average Drop (AD)** This metrics quantifies the answer to the question *"Did we hide in the image information which is relevant to the target class ?"*. To do so, the AD calculates the mean drop in classification confidence for the target class via the following formula :

$$\text{AD}(\%) = \frac{1}{N} \sum_{i=1}^{N} \frac{[p_i^c - o_i^c]_+}{p_i^c} \cdot 100. \tag{7}$$

- **Average Increase (AI)** This metric measures if the saliency map has removed unnecessary information from the input image, hence removing noise and improving the model's confidence in the target

class. This is done via the following formula :

$$\text{AI}(\%) = \frac{1}{N} \sum_{i=1}^{N} \mathbb{I}_{p_i^c < o_i^c} \cdot 100. \tag{8}$$

In the previous formulae, we denote by $N$ the size of the dataset and by $c$ the ground truth label of the input $i$. $p_i^c$ (respectively, $o_i^c$) represents the probability of classifying the image $i$ (respectively, the image $i$ masked by the inferred saliency map) in class $c$. We note that the AD and AI metrics are to be minimized and maximized respectively.

### 6.1.2 Self-supervised learning metrics

Since, to the best of our knowledge, no framework exists for the evaluation of saliency map based interpretability methods on self-supervised CNN models, we propose one. The proposed framework relies on four metrics; notably the Average Drop (AD), the Average Increase (AI) (Chattopadhay et al., 2018), the Mean Squared Error (MSE), and the mean Intersection over Union (mIoU) metric (Jaccard, 1901).

- **Average Drop (AD) and Average Increase (AI)** (Chattopadhay et al., 2018) To calculate the AD and AI, we estimate the saliency map from the self-supervised model, then evaluate the quality of the saliency map using pretrained supervised classification model and the formulae described in the previous Section 6.1.1. We detail the pretrained classification models used for each self-supervised model in Appendix B.4.

- **Mean Squared Error** Since self-supervised models output an encoding for each input image, we calculate the mean squared error between the encoding $z_i$ of the image $i$ and the encoding $\bar{z_i}$ of the image $i$ masked with the inferred saliency map. If the saliency map is able to isolate the important regions in the image which are encoded, there should be no difference between the encodings. This metric is to be minimized and we formulate it as follows :

$$MSE = \frac{1}{N} \sum_{i=1}^{N} ||z_i - \bar{z_i}||_2^2. \tag{9}$$

- **Mean Intersection over Union** (Jaccard, 1901) Under the hypothesis that the saliency map represents a segmentation highlighting the most important parts in an image, we calculate the mean Intersection over Union segmentation metric using the ground truth segmented images in the Pascal VOC dataset (Everingham et al., 2010). Formally, we calculate this metric using the following formula :

$$B = \begin{cases} 1 \text{ if saliency\_map } \leq T \\ 0 \text{ else} \end{cases} \qquad , \qquad \text{mIoU}(B,S) = \frac{|B \cap S|}{|B \cup S|}. \tag{10}$$

where $B$ is the pixel-wise binarized saliency map according to a threshold $T$, $S$ is the ground truth binary segmentation mask and $|B \cap S|$ (respectively, $|B \cup S|$) represents the area of intersection (respectively, union) between the saliency map and the ground truth segmentation $S$. Moreover, to be thorough, we study the impact of the performance of the different proposed interpretability methods as a function of the threshold parameter $T$ on a multitude of models. The obtained results are reported in Appendix C. We note that this metrics is to be maximized.

### 6.2 Results and discussion on supervised classification models

We evaluate and compare the TSM, MTSM and Multivec-EigenCAM method to a multitude of saliency map extraction methods on the pretrained VGG16 (Simonyan & Zisserman, 2014), Resnet50 (He et al., 2016) and ConvNext (Liu et al., 2022) models. For the evaluation, we calculate the AD and AI on the $50,000$ validation images of the ImageNet dataset. The obtained quantitative and qualitative results are presented in Table 1 and Figure 4 respectively.

Table 1: Saliency map methods comparison on the validation set of the ImageNet dataset across different pretrained classification models. The best values are highlighted in **bold**, while the second-best values are underlined. Cyan-colored rows denote label-independent saliency map methods.

| Method | Resnet50 AD(%) ↓ | Resnet50 AI(%) ↑ | ConvNext AD(%) ↓ | ConvNext AI(%) ↑ | VGG16 AD(%) ↓ | VGG16 AI(%) ↑ |
|---|---|---|---|---|---|---|
| GradCAM | **13.01** | **44.52** | 32.49 | 14.67 | 17.58 | 40.32 |
| GradCAM++ | 13.61 | 42.50 | 80.32 | 9.72 | 19.12 | 35.20 |
| HiResCAM | **13.01** | **44.52** | 32.49 | 16.66 | 19.18 | 37.59 |
| XGradCAM | **13.01** | **44.52** | 32.49 | 16.66 | 16.00 | **41.64** |
| AblationCAM | 13.21 | 43.05 | 51.74 | 10.99 | 17.66 | 38.72 |
| OptiCAM | 26.45 | 27.77 | **19.60** | **26.77** | **15.59** | 40.48 |
| DeepLIFT | 95.37 | 2.50 | 93.44 | 1.75 | 95.34 | 2.47 |
| DeepSHAP | 95.36 | 2.49 | 93.43 | 1.75 | 95.33 | 2.47 |
| LIME | 57.73 | 12.71 | 56.23 | 10.92 | 62.33 | 11.47 |
| EigenCAM | 38.51 | 21.41 | 36.24 | 9.69 | 47.32 | 17.11 |
| TSM | 16.86 | 37.27 | 30.85 | 9.36 | 26.74 | 27.50 |
| Multivec-EigenCAM | 22.69 | 30.43 | 37.93 | 9.92 | 25.64 | 27.50 |
| MTSM | 16.27 | 37.24 | 31.78 | 9.16 | 22.02 | 30.65 |

From Table 1, we notice that CAM methods' performance is superior to that of feature attribution methods as they highlight larger contiguous regions important to the classification task. Moreover, when comparing label-dependent CAM methods to our proposed label-independent methods, we notice that the latter methods achieve competitive performance especially, on the ConvNext model where we note TSM's second best result on the AD metric. However, label-independent methods' performance is still lower than state-of-the-art label-dependent methods on classification models as the latter utilize more information. A fairer comparison of our proposed methods is with label-independent methods notably EigenCAM. In this setting, TSM demonstrates approximately a 50% improvement in performance across most models, measured by the AD and AI metrics. For example, on the VGG16 model, TSM achieves a 26.96% Average Drop compared to the 47.32% obtained by the EigenCAM method, representing a 56.97% improvement. Additionally, Multivec-EigenCAM and MTSM improve upon the performance of EigenCAM and TSM respectively. The magnitude of the difference in performance between uni-vector and multi-vector methods is governed by the amount of variance encoded in the singular vectors following the first one as discussed in Section 5. We perceive a larger performance jump going from EigenCAM to Multivec-EigenCAM compared to going from TSM to MTSM. This goes back to the more accurate estimation of the singular vectors by Tucker tensor decomposition and capturing of most of the variance in the first singular vector. Therefore, even though MTSM results in a smaller performance improvement gap, it significantly improves upon the performance of Multivec-EigenCAM. We further emphasize the importance of using the Tucker tensor decomposition instead of the SVD as we notice that on two out of the three tested models the uni-vector TSM achieves significantly better performance than Multivec-EigenCAM.

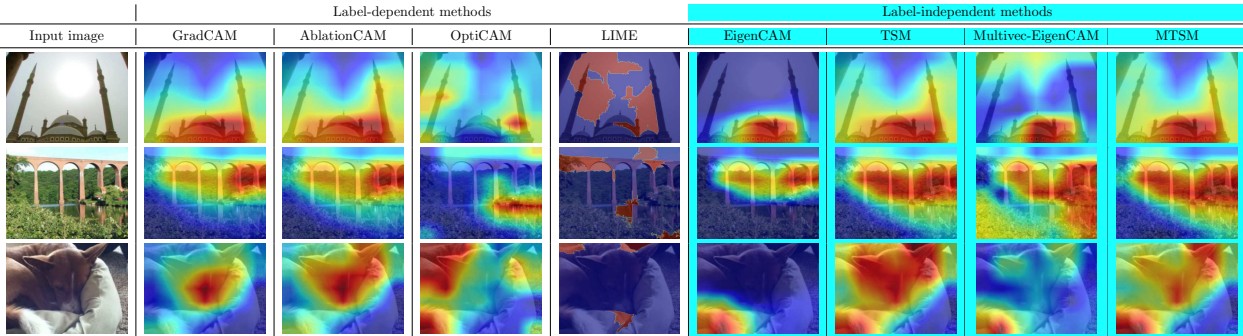

Figure 4: Qualitative comparison of saliency map-based methods on the Resnet50 model.

Figure 4 presents a qualitative comparison between a sample of label-dependent and independent methods (Additional figures are presented in Appendix E). This figure backs our quantitative results where we notice that TSM and MTSM yield saliency maps similar to the most performant label-dependent method notably GradCAM. However, the highlighted regions by TSM and MTSM can be relevant to the image class or not, showcasing the label-independent nature of the methods. This is particularly noticeable in the second image, where TSM and MTSM effectively highlights the bridge and the river instead of just the bridge as in the label-dependent methods. Furthermore, compared to EigenCAM and Multivec-EigenCAM, TSM and MTSM highlight more of the regions in the image and with higher confidence (Notice the mosque and dog images in Figure 4). Finally, comparing uni-vector and multi-vector, we perceive a significant improvement of Multivec-EigenCAM over EigenCAM with more highlighted regions. Nevertheless, between TSM and MTSM little difference can be observed which joins our observation and discussion of the quanlitative results.

### 6.3 Results and discussion on self-supervised models

In this section, we present the quantitative and qualitative results obtained by the methods we propose. We analyse, discuss and compare them only to the results obtained by the EigenCAM method (Muhammad & Yeasin, 2020) as it is the only principled method which can be applied in this context[3]. In this section, we calculate the AD, AI and MSE metrics on the $50,000$ validation images of the ImageNet dataset (Deng et al., 2009) whereas the mIoU metric is calculated on the $1,449$ validation images of the Pascal VOC dataset (Everingham et al., 2010). The obtained quantitative and qualitative results are presented in Table 2 and Figures 5, and 6.

Table 2: Comparison between TSM, EigenCAM, MTSM and Multivec-EigenCAM on self-supervised models trained on the ImageNet dataset. The reported mIoU values are computed with a threshold of 0.5 applied to the saliency maps. The best values are highlighted in **bold**.

| Method | EigenCAM | | | | TSM | | | |
|---|---|---|---|---|---|---|---|---|
| | AD(%) ↓ | AI(%) ↑ | MSE×$10^{-3}$ ↓ | mIoU(%) ↑ | AD(%) ↓ | AI(%) ↑ | MSE×$10^{-3}$ ↓ | mIoU(%) ↑ |
| Moco V2 | 38.42 | 32.56 | 0.25 | 23.13 | **20.98** | **44.60** | **0.16** | **28.82** |
| SWAV | 31.94 | 37.49 | 12.22 | 26.35 | **14.61** | **47.23** | **5.55** | **39.97** |
| Barlow Twins | 41.72 | 31.02 | 8.73 | 21.06 | **21.41** | **44.69** | **5.95** | **27.91** |
| VicRegL Resnet50 | 42.15 | 30.80 | 75.50 | 21.28 | **20.30** | **45.36** | **48.20** | **28.47** |
| VicRegL ConvNext | 40.71 | 9.93 | 95.54 | 22.65 | **17.42** | **13.26** | **25.57** | **36.69** |

| Method | Multivec-EigenCAM | | | | MTSM | | | |
|---|---|---|---|---|---|---|---|---|
| | AD(%) ↓ | AI(%) ↑ | MSE×$10^{-3}$ ↓ | mIoU(%) ↑ | AD(%) ↓ | AI(%) ↑ | MSE×$10^{-3}$ ↓ | mIoU(%) ↑ |
| Moco V2 | 19.24 | 45.49 | 0.15 | 27.75 | **16.38** | **46.96** | **0.12** | **33.57** |
| SWAV | 16.40 | 44.81 | 6.00 | 37.54 | **13.89** | **46.59** | **4.63** | **41.33** |
| Barlow Twins | 18.96 | 45.84 | 5.26 | 32.35 | **16.58** | **46.42** | **4.18** | **36.15** |
| VicRegL Resnet50 | 18.10 | 45.59 | 42.37 | 33.19 | **16.30** | **46.67** | **35.98** | **35.38** |
| VicRegL ConvNext | **17.10** | 15.85 | 31.49 | 26.61 | 17.87 | **13.51** | **26.84** | **37.00** |

From Table 2, we observe the superior performance of Tucker decomposition based saliency map methods (i.e. TSM and MTSM) compared to the SVD based saliency map methods (i.e. EigenCAM and Multivec-EigenCAM). Additionally, similar to our observation on the supervised classification model, TSM acheives a 50% improvement in performance on all metrics compared to EigenCAM. This is attributed to the benefits of using the most suitable mathematical tools to manipulate tensors notably the Tucker tensor decomposition. Also, the use of the absolute value in the TSM and MTSM methods brings forward negative contributions which are of interest. Consequently, through the used metrics we can conclude that compared to EigenCAM, TSM is able to :

- Retain more of crucial information in the image which is relevant to image classification.
- Produce saliency maps aligned with segmentation masks.

---

[3]Label-dependent methods like GradCAM, which we test in the previous section, are difficult to adapt to this setting in a principled manner and can represent an interesting future work (See our discussion in Section 3.2.2).

- Conceal notably less input regions vital to the self-supervised model.

Additionally, we notice that the multi-vector methods improve on the performance of TSM. Also, Multivec-EigenCAM has a significant performance jump compared to EigenCAM as opposed to the performance gap between TSM and MTSM. This can also be explained by our previous argument on the supervised classification models which links back to the amount of variance encoded by the first singular vectors. However, for the VicRegL ConvNext model we notice a slight drop in performance going from TSM to MTSM. The reason for this is illustrated in Figure 3(a). We can see that a large portion the variance is captured by the first singular vector around 30% to 40%. Consequently, the rest of the singular values do not bring significant information. Combined with the weight assigned to these vectors, their impact on the final saliency map is minimal at best or adds noise to it.

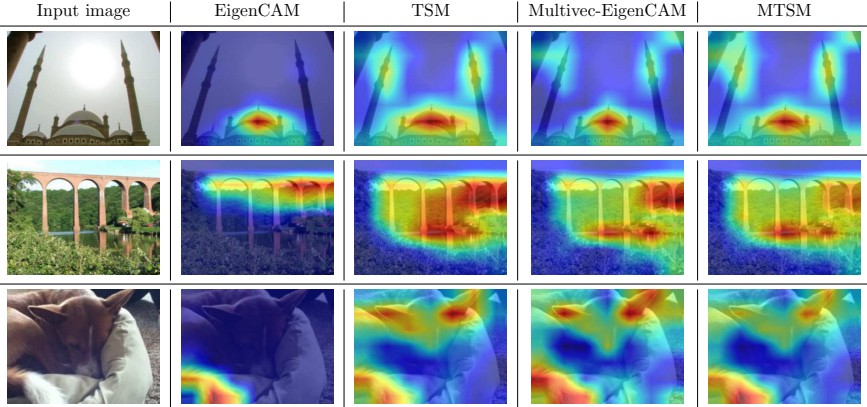

Figure 5: Qualitative comparison of saliency maps extracted using the EigenCAM versus TSM on the SWAV model applied to the ImageNet dataset.

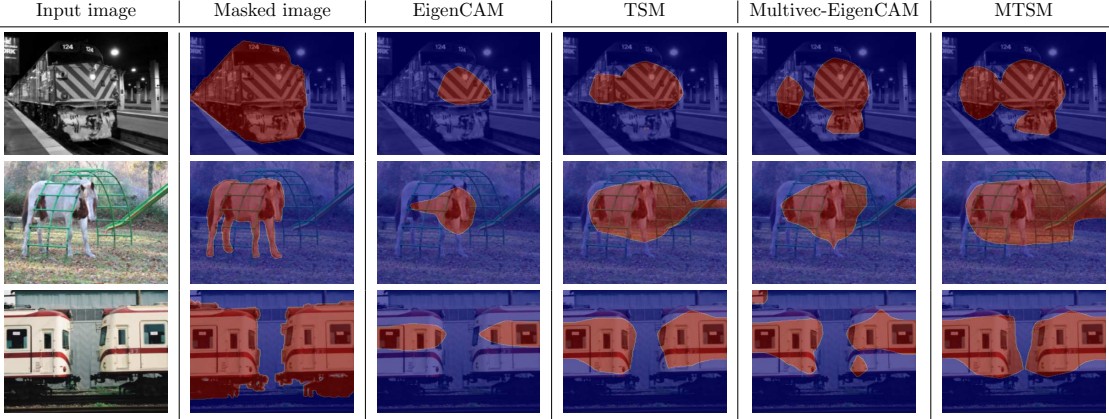

Figure 6: Qualitative comparison of saliency maps extracted using the EigenCAM versus TSM on the SWAV model, juxtaposed with the segmentation mask from the Pascal VOC dataset. The showcased segmentations based on the saliency maps employ a threshold of 0.5.

Figures 5, and 6 qualitatively confirm the quantitative results of Table 2. These figures illustrate a superior coverage of important objects in the saliency maps generated by TSM compared to those produced by EigenCAM. For instance, we notice TSM providing better coverage of distinctive features in the dog and the mosque images. Lastly, the visual representations in Figure 6 underscore the qualitative findings reflected in the mIoU metric of Table 2. Notably, EigenCAM (respectively Multivec-EigenCAM) highlights fewer regions of interest than TSM (respectively MTSM), leading to a lower mIoU. Moreover, in Figure 5, EigenCAM presents a substancial difference in the resulting saliency map compared to Multivec-EigenCAM whereas the latter produces saliency maps inline with those of TSM and MTSM. Consequently, EigenCAM is not able to capture all the necessary information and presents a very limited view of what the self-supervised model is

focusing on. Therefore, when using EigenCAM as an interpretability method the conclusion that one could arrive to might not be the correct one.

Lastly, we compare the saliency maps produced through supervised classification models in Figure 4 and the ones produced through self-supervised models in Figure 5. We notice that all the label-independent saliency map methods are not class specific but the learning task has an impact on the produced map. This is particularly evident on the first and third images.

Table 3: Execution time comparison between label-independent methods in seconds to produce a single saliency map for a single input image. In orange we outline the difference between the SVD methods and Tucker decomposition methods.

| Model | EigenCAM | TSM | Multivec-EigenCAM | MTSM |
|---|---|---|---|---|
| Resnet50 | 1.64 | 3.68 (2.2×) | 1.90 | 4.05 (2.1×) |
| ConvNext | 1.19 | 1.65 (1.4×) | 1.14 | 1.73 (1.5×) |
| VGG16 | 1.10 | 1.21 (1.1×) | 1.12 | 1.22 (1.1×) |
| Moco V2 | 1.13 | 2.64 (2.3×) | 1.12 | 2.85 (2.5×) |
| SWAV | 1.15 | 2.35 (2.0×) | 1.15 | 2.56 (2.2×) |
| Barlow Twins | 1.17 | 3.32 (2.8×) | 1.17 | 3.62 (3.1×) |
| VicRegL Resnet50 | 1.22 | 2.48 (2.1×) | 1.19 | 2.53 (2.1×) |
| VicRegL ConvNext | 0.99 | 1.23 (1.2×) | 0.99 | 1.26 (1.3×) |

From Table 3 we notice that Tucker decomposition methods even though they output high quality saliency maps they come with an increased computational cost compared to SVD based methods. The former methods' mean processing time is approximately two times slower. However, going from uni-vector methods to multivector methods come with a negligible runtime increase.

# 7 Conclusion

In this work, we introduce the TSM method, a label-independent CNN interpretability method harnessing tensor algebra notably Tucker tensor decomposition to produce high quality and fidelity saliency maps of CNN's input regions. TSM is a principled method, not restricted to supervised classification CNN models and yet achieves competitive quantitative and qualitative results compared to label-dependent methods. Moreover, compared to the only CNN label-independent method namely EigenCAM, TSM allows for a more accurate estimation of singular vectors and values resulting in more accurate saliency maps. Furthermore, we extend the EigenCAM and TSM method to the Multivec-EigenCAM and MTSM which take profit of all the singular vectors and values further improving the interpretability of supervised and self-supervised CNN models. We test all the label-independent methods on various supervised classification CNN models via established metrics and on self-supervised CNN models through an evaluation procedure we propose. When comparing EigenCAM to TSM, our results indicate a 50% improvement in quantitative results on all tested metrics and models. Multivec-EigenCAM and MTSM further improve on the performance of their base methods with varying extents depending on the amount of variance encoded by each singular vector. Our qualitative results showcase that EigenCAM can be misleading when used as an interpretability method. This is because it highlights fewer regions of the input in the saliency map compared to our proposed methods which are more reliable since they highlight all the significant regions.

It worth noting that TSM and MTSM come with increased computational costs, requiring approximately 2× more processing time than EigenCAM and Multivec-EigenCAM respectively due to the Tucker tensor decomposition. Despite this complexity, the significant improvements in interpretability and the methods' effectiveness in interpreting self-supervised models justify the additional computational burden. Future work will focus on optimization techniques to reduce execution time, ensuring that these methods remain efficient. Moreover, in this work we have tailored the TSM and MTSM methods to particularly interpret CNN models. However, in light of their state-of-the-art performance for self-supervised interpretability on CNN models, an interesting perspective work is to explore innovative ways to utilize tensor algebra to better interpret other neural network architectures such as transformers.

**Acknowledgements**

The authors extend their gratitude to Stéphane Ayache and Rachid Deriche for their invaluable input and constructive feedback on the content of this paper. This work received access to the High-Performance Computing (HPC) resources of MesoPSL, financed by the Region Île-de-France and the Equip@Meso project (reference ANR-10-EQPX-29-01) of the Investissements d'avenir program supervised by the Agence nationale pour la recherche.

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

# Supplementary Material

## A Code

Hereafter, we present the implementation of the TSM, Multivec-EigenCAM en MTSM methods proposed in this work using the PyTorch library for Class Activation Map methods, as referenced in Jacob Gildenblat et al. (2021).

```python
import torch
import numpy as np
from pytorch_grad_cam.base_cam import BaseCAM
from tensorly.decomposition import tucker

class TSM(BaseCAM):
   def __init__(self, model, target_layers, use_cuda=False,  reshape_transform=None):
       super(TSM, self).__init__(model, target_layers,  use_cuda, reshape_transform,
                                             uses_gradients=False)

   def get_cam_image(self, input_tensor, target_layer,  target_category, activation_batch,
                                           grads, eigen_smooth):
       feature_maps = torch.from_numpy(activation_batch)
       VT = [tucker(feature_map.numpy(), rank=list(feature_map.shape))[1][0] for
                                           feature_map in feature_maps]
       VT = torch.from_numpy(np.array(VT))

       VT = VT[:,0].unsqueeze(-1).unsqueeze(-1)
       S = (feature_maps * VT).sum(dim=1)
       return S.abs().numpy()
```

```python
import torch
import numpy as np
from pytorch_grad_cam.base_cam import BaseCAM

class MultiVecEigenCam(BaseCAM):
  def __init__(self, model, target_layers, use_cuda=False, reshape_transform=None):
    super().__init__(model=model, target_layers=target_layers, reshape_transform=
                                        reshape_transform, uses_gradients=False)

def get_cam_image(self, input_tensor, target_layer, target_category, activation_batch,
                                      grads, eigen_smooth):
  size = (activation_batch.shape[0], activation_batch.shape[-2], activation_batch.shape[-1
                                      ])
  feature_maps = torch.from_numpy(activation_batch)
  feature_maps = feature_maps.view(
  feature_maps.shape[0], feature_maps.shape[1], -1).permute(0, 2, 1)
  feature_maps = feature_maps - feature_maps.mean(dim=1, keepdim=True)

  _, sigma, VT = torch.linalg.svd(feature_maps, full_matrices=True)
  sigma = sigma.real
  sigma = sigma/sigma.max(dim=1, keepdim=True).values

  cams = []
  for i in range(sigma.shape[1]):
    tmp = VT[:, i].unsqueeze(-1)
    projection = torch.bmm(feature_maps, tmp).view(size)
    weight = sigma[:, i, None, None]
    projection = projection * weight
    cams.append(projection.abs().numpy())

  cam = np.array(cams)
  return cam.mean(0)
```

```python
import torch
import numpy as np
from pytorch_grad_cam.base_cam import BaseCAM
from tensorly.decomposition import tucker

class MTSM(BaseCAM):
  def __init__(self, model, target_layers, use_cuda=False, reshape_transform=None):
    super(MTSM, self).__init__(model=model, target_layers=target_layers, reshape_transform
                                        =reshape_transform, uses_gradients=False)

  def get_cam_image(self, input_tensor, target_layer, target_category, activation_batch,
                                        grads, eigen_smooth):
    feature_maps = torch.from_numpy(activation_batch)
    VT = []
    singular_values = []
    for feature_map in feature_maps:
    t = feature_map.numpy()
    core, factors = tucker(t, rank=list(t.shape))
    VT.append(factors[0])
    singular_values.append(np.linalg.norm(core, axis=(-2,-1), ord='fro'))

    VT = torch.from_numpy(np.array(VT))
    singular_values = torch.from_numpy(np.array(singular_values))
    singular_values /= singular_values.max(dim=1, keepdim=True).values

    saliency_maps = []
    for i in range(singular_values.shape[1]):
      tmp = VT[:,i].unsqueeze(-1).unsqueeze(-1)
      projection = (feature_maps * tmp).mean(dim=1)
      weight = singular_values[:,i].unsqueeze(-1).unsqueeze(-1)
      projection = projection * weight
      saliency_maps.append(projection.detach().abs().numpy())

    S = np.array(saliency_maps)
    S = np.transpose(S, axes=(1,0,2,3))
    S = S.mean(1)
    return S
```

## B  Experimental setup

### B.1  Selected layers per model for the CAM inference

In Table 4, we outline the chosen target layers for each examined model. These layers serve as the source from which we extract the feature map tensor for computing the saliency map.

Table 4: Designated layers for saliency map calculation across all tested models and CAM methods in this paper.

| Model | | Layer |
|---|---|---|
| Resnet50 | (He et al., 2016) | model.layer4[-1] |
| ConvNext | (Liu et al., 2022) | model.features |
| VGG16 | (Simonyan & Zisserman, 2014) | model.features |
| Moco V2 | (Chen et al., 2020b) | model.layer4[-1] |
| SWAV | (Caron et al., 2020) | model.layer4[-1] |
| Barlow Twins | (Zbontar et al., 2021) | model.layer4[-1] |
| VicRegL Resnet50 | (Bardes et al., 2022) | model.layer4[-1] |
| VicRegL ConvNext | (Bardes et al., 2022) | model.stages[3][2].dwconv |

### B.2 Datasets

We evaluate the different saliency map calculation methods in this work on the ImageNet 2012 (Deng et al., 2009) and Pascal VOC (Everingham et al., 2010) datasets. All images of these datasets have been rescaled to a size of $224 \times 224$, and the values of the pixels have been normalized to match the preprocessing of the used pre-trained CNNs.

**ImageNet 2012** For this dataset, we use the $50,000$ validation images of the ImageNet ILSVRC 2012 dataset (Deng et al., 2009). We mainly use this dataset to calculate the Average Drop, Average Increase and Mean Squared Error metrics.

**Pascal VOC** Owing to the availability of the segmentation masks on the Pascal VOC 2012 challenge dataset (Everingham et al., 2010), we harness this dataset to calculate the mean Intersection over Union metric (Jaccard, 1901).

### B.3 Used libraries

The implementation of the different CAM based saliency map methods is done via the *PyTorch library for CAM methods* (Jacob Gildenblat et al., 2021), the DeepLIFT and DeepSHAP methods are implemented using the Captum Python library (Kokhlikyan et al., 2020) and the LIME method is imported from the official implementation in (Ribeiro et al., 2016).

### B.4 Used classification pretrained models

In Table 5, we provide a list of pretrained supervised classification models utilized for computing the Average Drop and Average Increase metrics on the self-supervised models. The primary rationale behind selecting these classification models is their alignment with the backbone model employed in the self-supervised models.

Table 5: Utilized supervised classification models to calculate the metrics of Average Drop and Average Increase on the self-supervised models.

| Self-supervised model | | Classification model | |
|---|---|---|---|
| Moco V2 | (Chen et al., 2020b) | Resnet50 | (He et al., 2016) |
| SWAV | (Caron et al., 2020) | Resnet50 | (He et al., 2016) |
| Barlow Twins | (Zbontar et al., 2021) | Resnet50 | (He et al., 2016) |
| VicRegL Resnet50 | (Bardes et al., 2022) | Resnet50 | (He et al., 2016) |
| VicRegL ConvNext | (Bardes et al., 2022) | ConvNext | (Liu et al., 2022) |

## C Hyperparameter study for the mIoU metric on self-supervised models

In Figure 7, we illustrate our investigation into the threshold hyperparameter employed to binarize saliency maps into segmentation masks and its impact on the mIoU metric. We systematically vary the binarization threshold from 0.4 to 0.9 with a step size of 0.1 and observe the corresponding mIoU values. We exclude threshold values lower than 0.4 as we deem them irrelevant for segmentation. From the observations in Figure 7, it becomes evident that, irrespective of the pretrained self-supervised model, TSM and MTSM yield a more accurate approximation of the segmentation mask than EigenCAM or Multivec-EigenCAM.

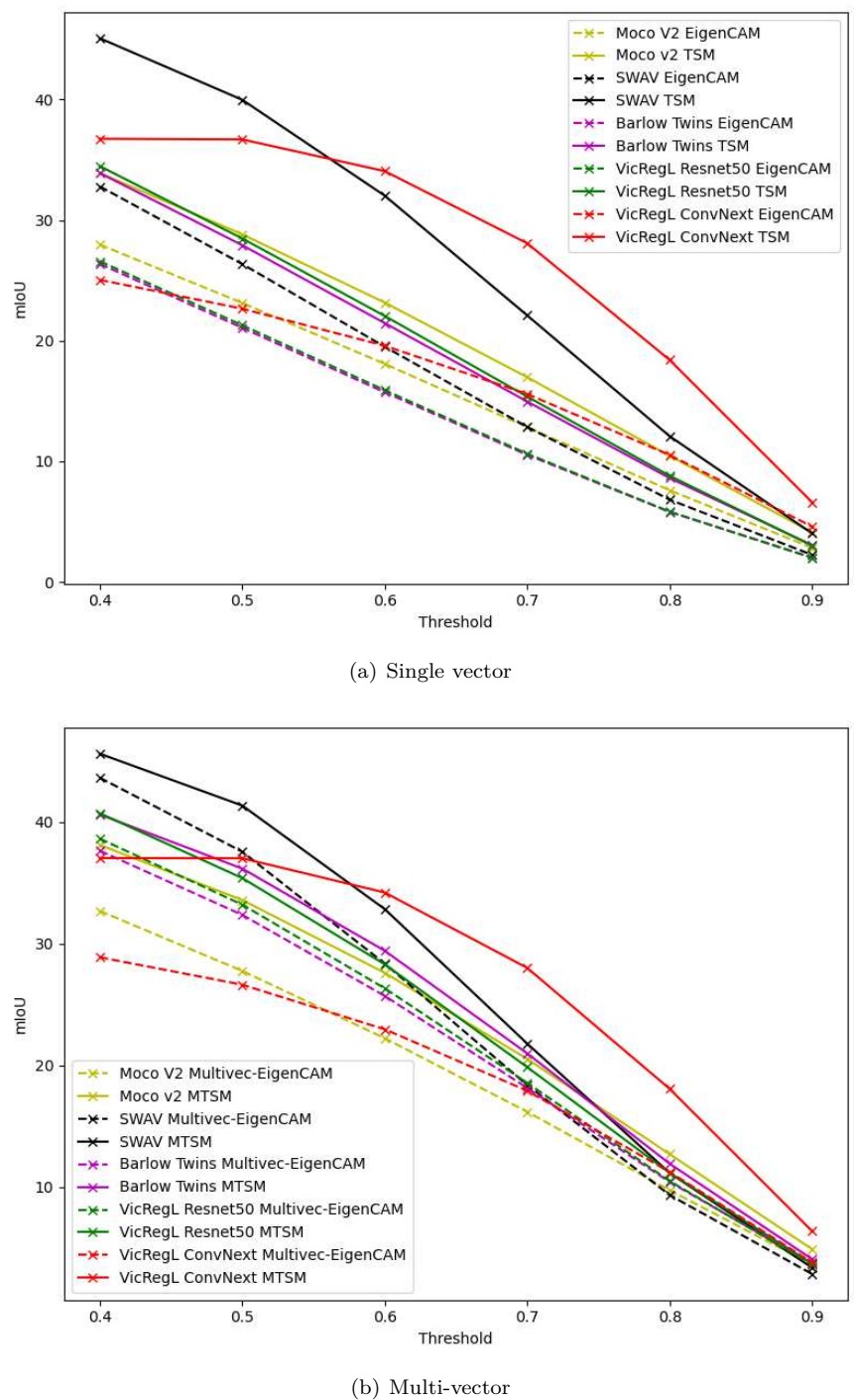

(a) Single vector

(b) Multi-vector

Figure 7: On the exploration of the threshold hyperparameter for Class Activation Map (CAM) binarization and its influence on the mIoU metric, solid lines in the plot depict the mIoU values obtained using Tucker decomposition based CAM methods, while dashed lines represent the mIoU values obtained using SVD based CAM methods.

## D    Additional self-supervised models' saliency maps

From the first three images in the Figures 8, 9, 10, 11, 12, 13, 14 and 15 we notice that when we have one class in the image with multiple subjects all the saliency map methods are able to identify the objects in the input image. However, when we have two classes, as in the forth image with the dog and the cat at the top and bottom of the image, TSM outperforms EigenCAM as it is able to locate the cat in the image which EigenCAM fails to do. Furthermore, all the multivector methods are able to identify all the objects in the image.

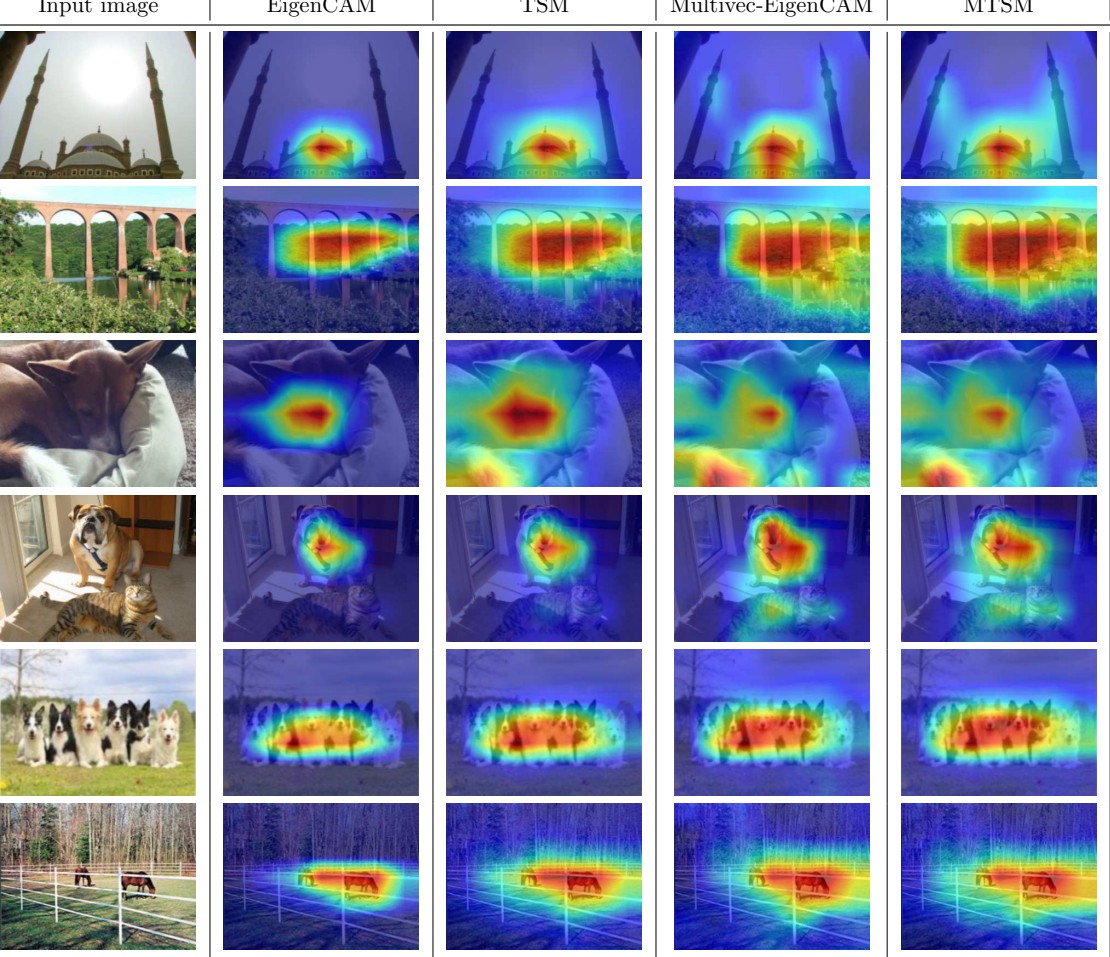

Figure 8: Qualitative comparison of the saliency maps extracted using the EigenCAM method versus the TSM, Multivec-EigenCAM and MTSM methods on the VicRegL Resnet50 model (Bardes et al., 2022) applied to the ImageNet dataset (Deng et al., 2009).

| Input image | EigenCAM | TSM | Multivec-EigenCAM | MTSM |
|---|---|---|---|---|

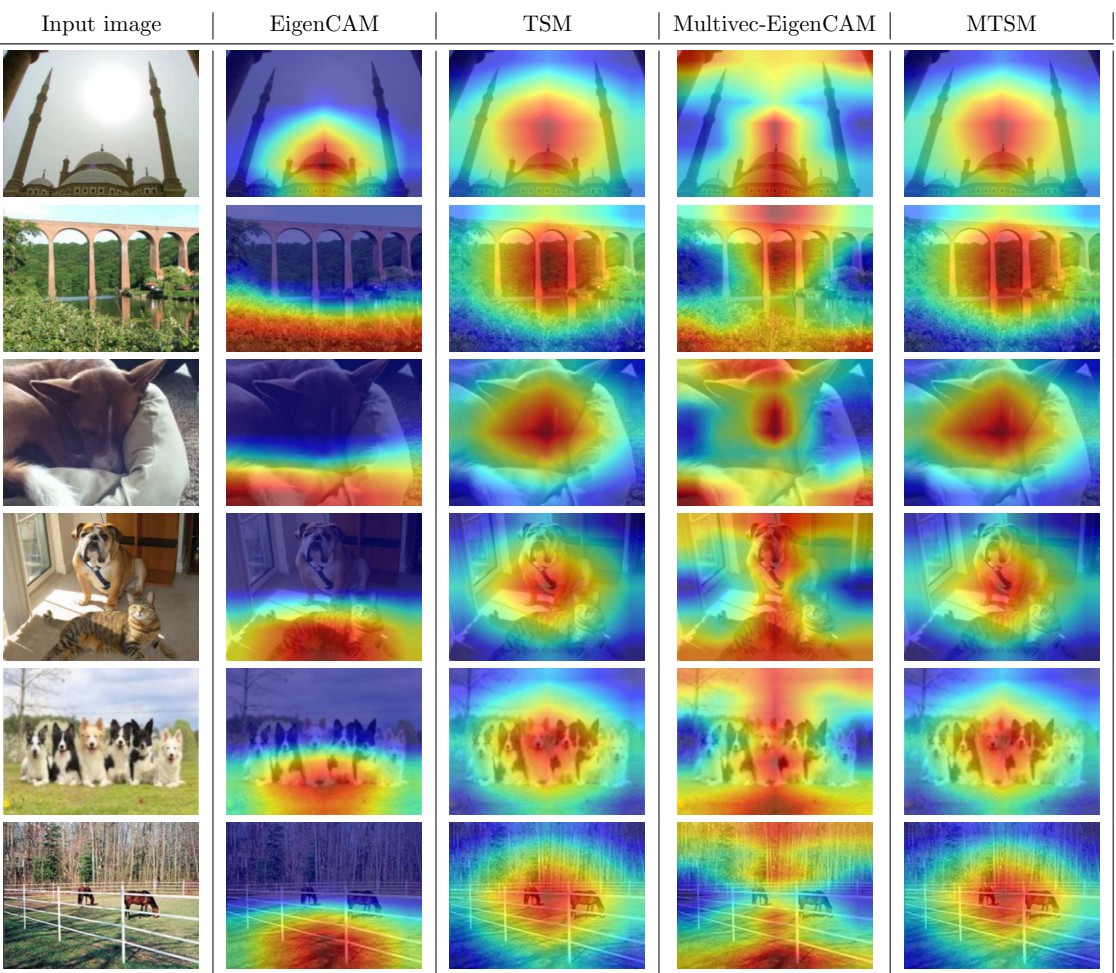

Figure 9: Qualitative comparison of the saliency maps extracted using the EigenCAM method versus the TSM, Multivec-EigenCAM and MTSM methods on the VicRegL ConvNext model (Bardes et al., 2022) applied to the ImageNet dataset (Deng et al., 2009).

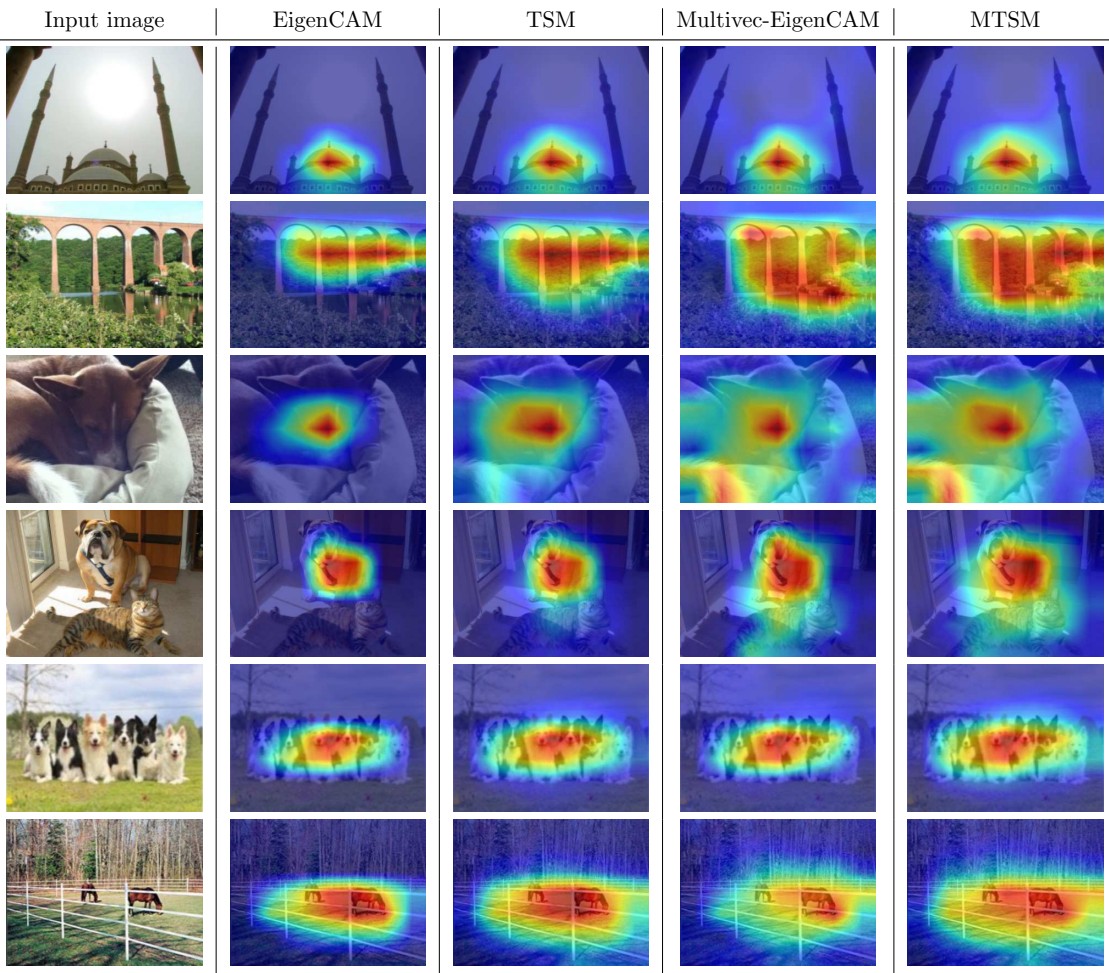

Figure 10: Qualitative comparison of the saliency maps extracted using the EigenCAM method versus the TSM, Multivec-EigenCAM and MTSM methods on the Barlow Twins model (Zbontar et al., 2021) applied to the ImageNet dataset (Deng et al., 2009).

| Input image | EigenCAM | TSM | Multivec-EigenCAM | MTSM |
|---|---|---|---|---|

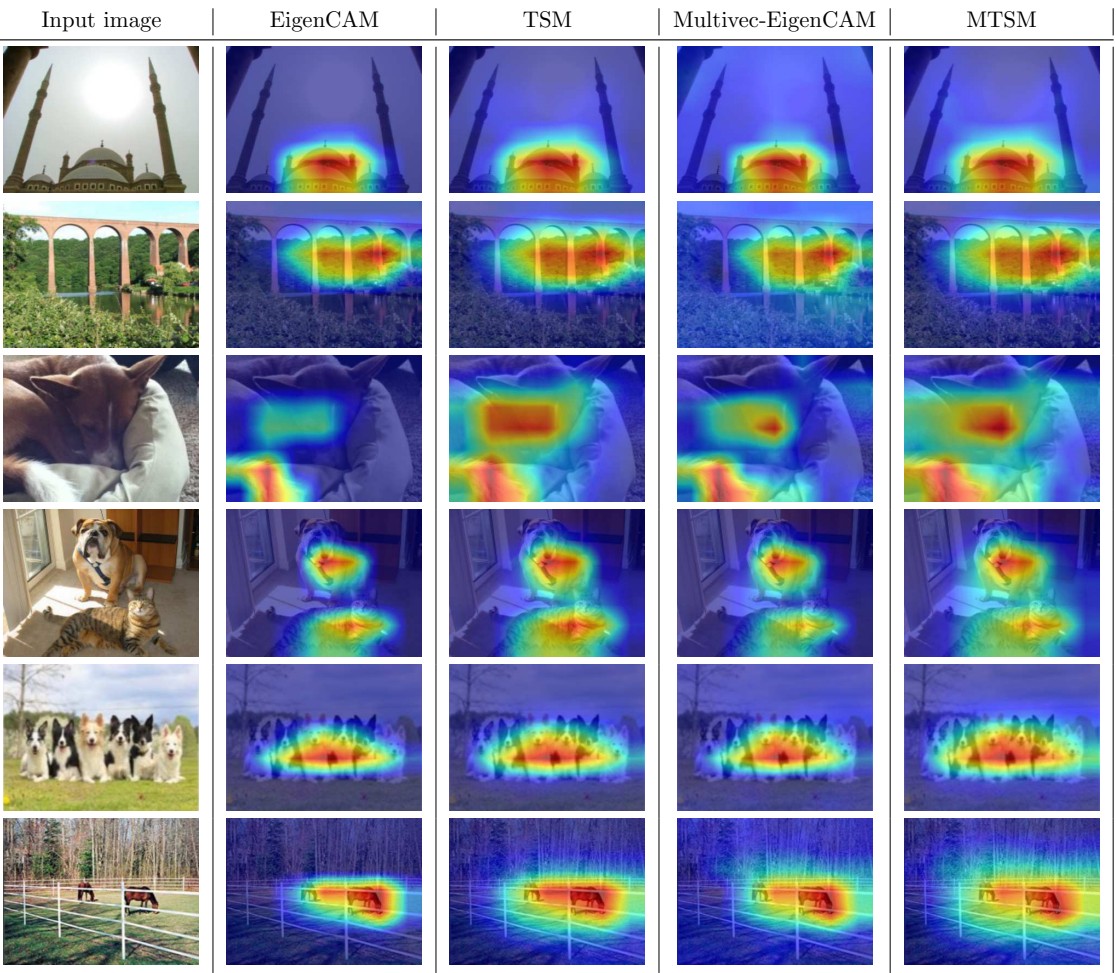

Figure 11: Qualitative comparison of the saliency maps extracted using the EigenCAM method versus the TSM, Multivec-EigenCAM and MTSM methods on the Moco V2 model (Chen et al., 2020b) applied to the ImageNet dataset (Deng et al., 2009).

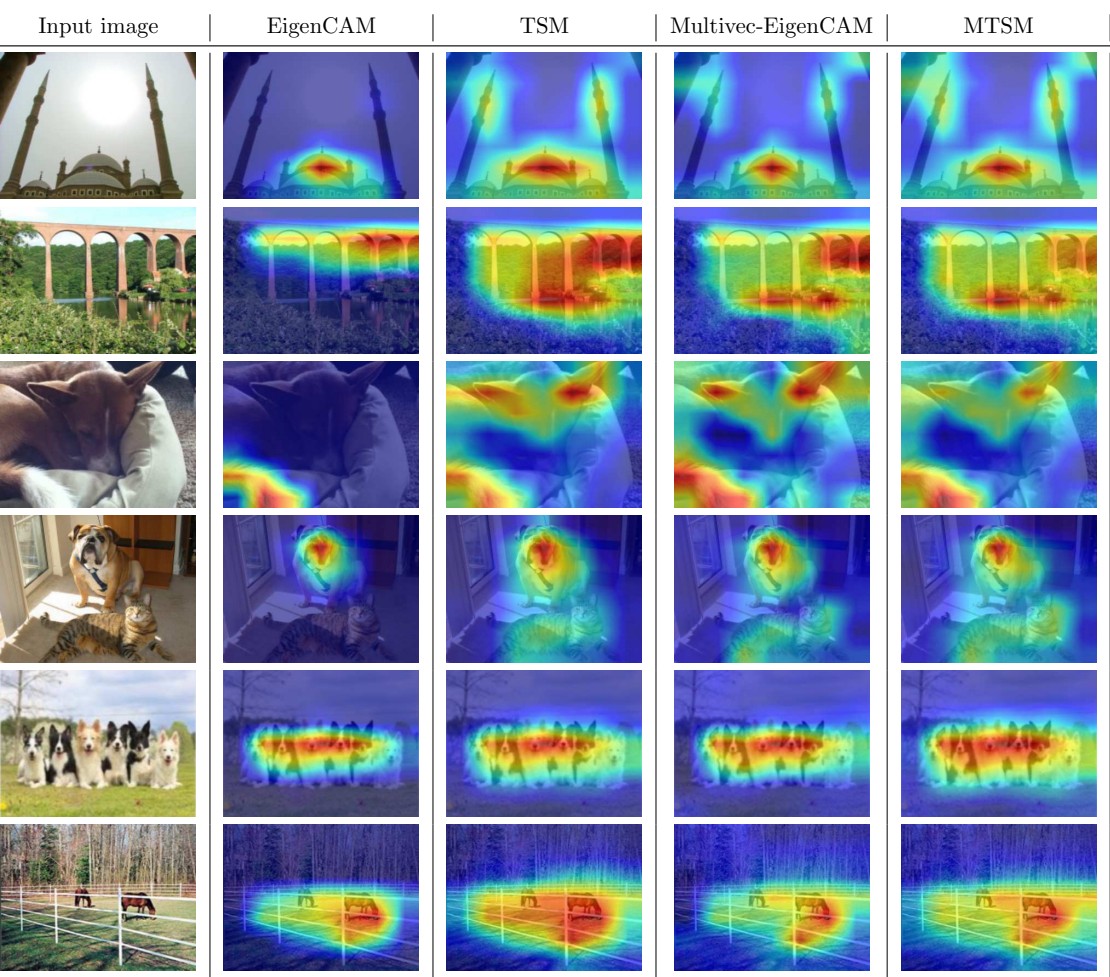

Figure 12: Qualitative comparison of the saliency maps extracted using the EigenCAM method versus the TSM, Multivec-EigenCAM and MTSM methods on the SWAV model (Caron et al., 2020) applied to the ImageNet dataset (Deng et al., 2009).

## E    Additional supervised models' saliency maps

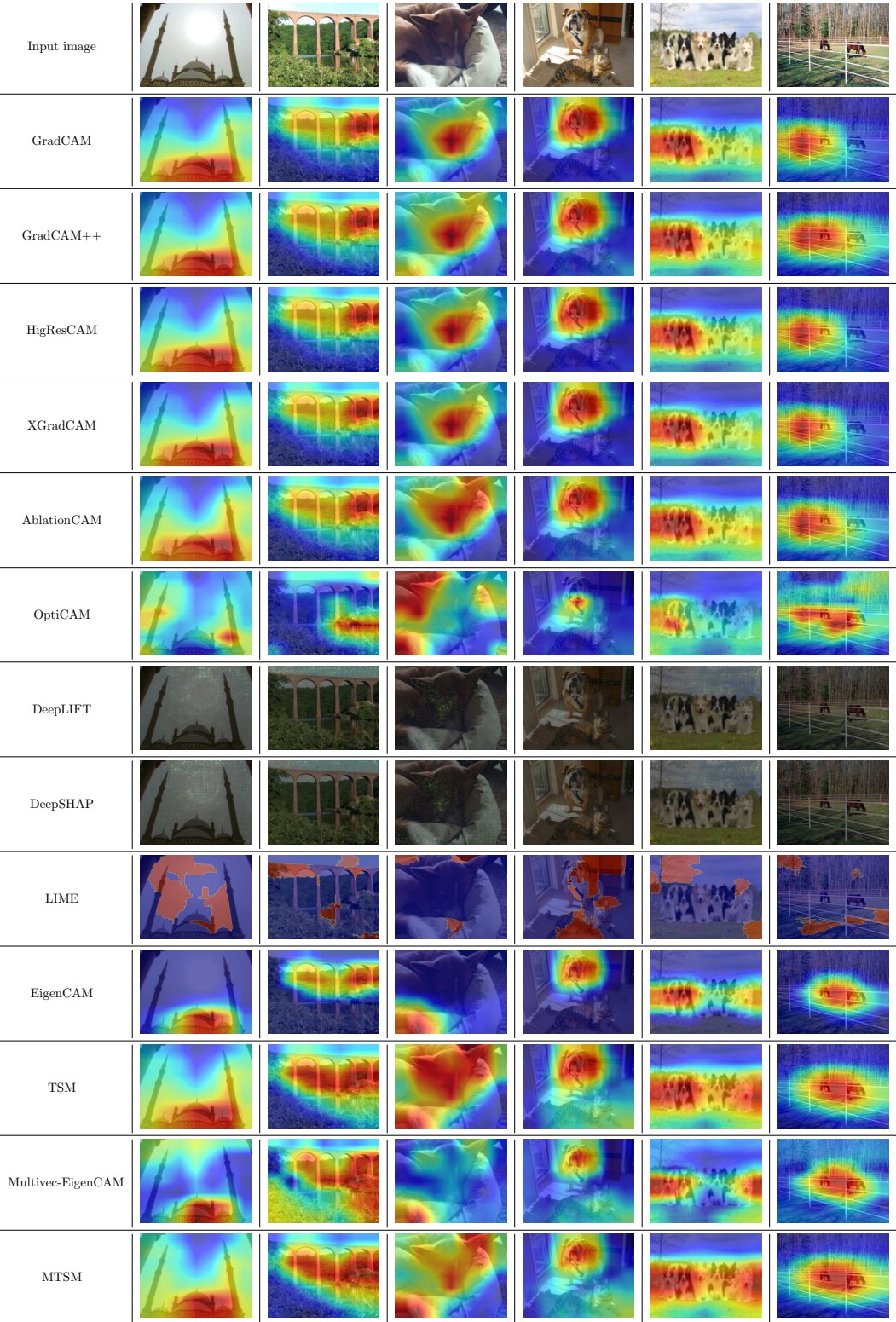

Figure 13: Qualitative comparison of saliency maps calculation methods on the supervised classification Resnet50 model (He et al., 2016) applied to the ImageNet dataset (Deng et al., 2009).

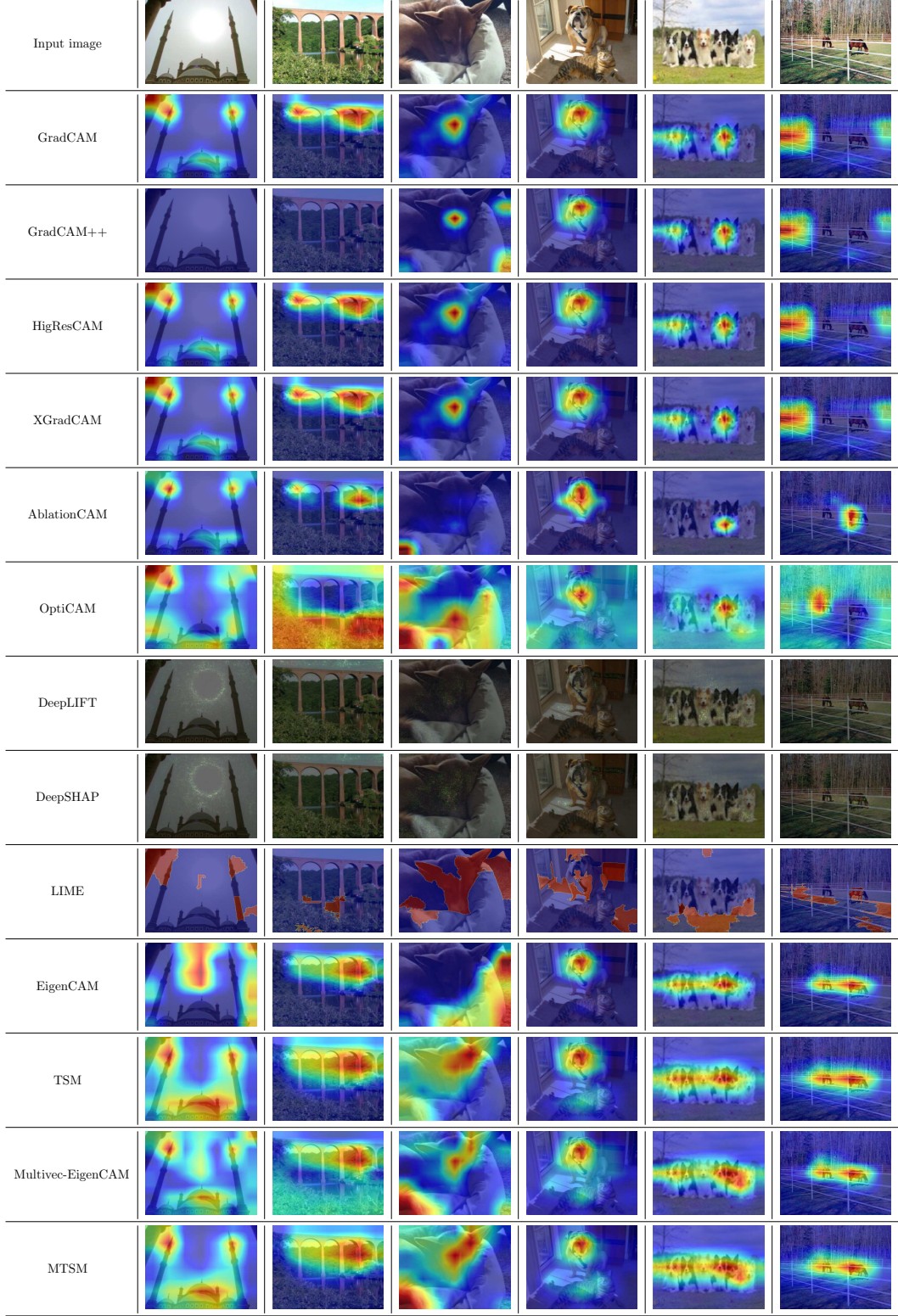

Figure 14: Qualitative comparison of saliency maps calculation methods on the supervised classification ConvNext model (Liu et al., 2022) applied to the ImageNet dataset (Deng et al., 2009).

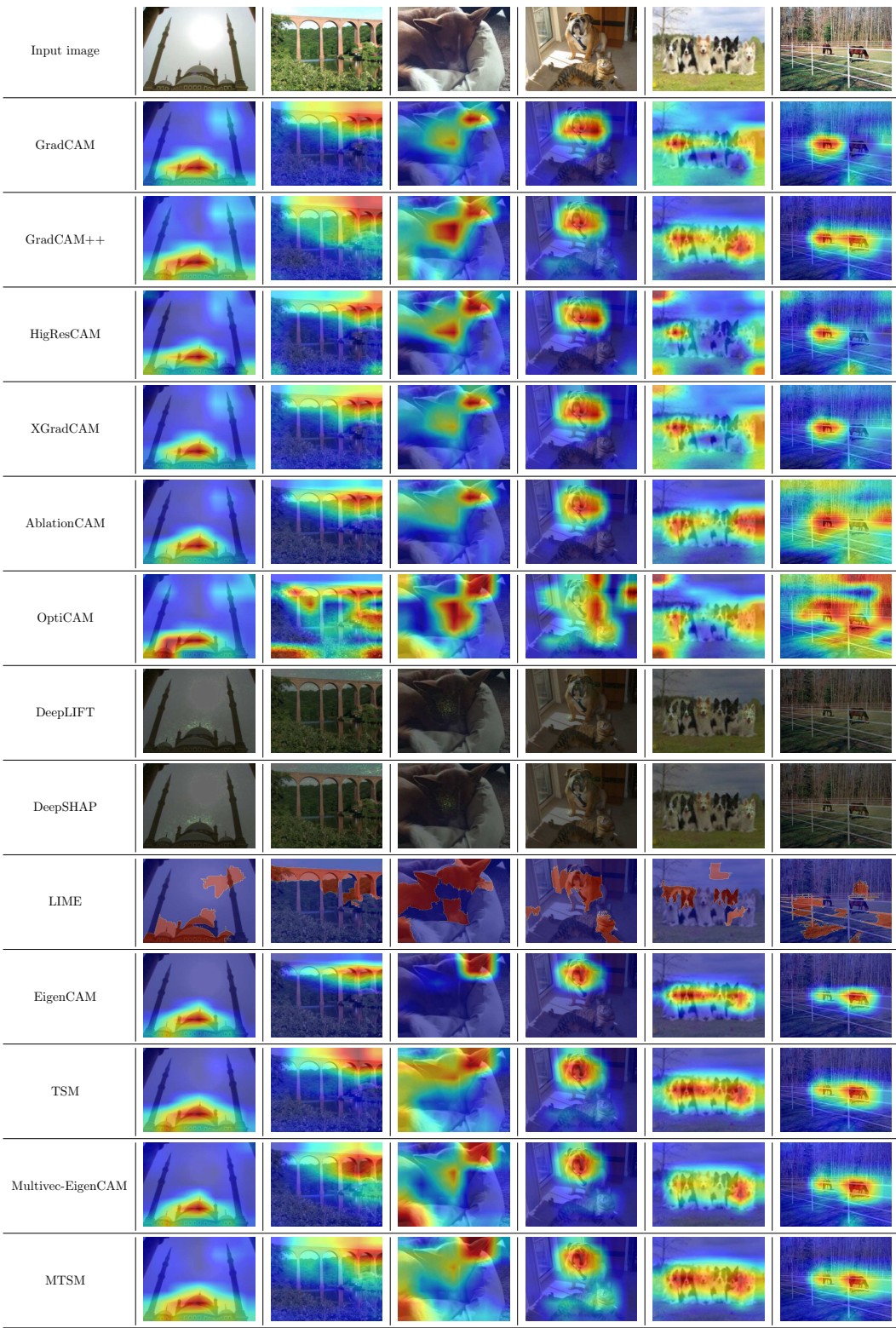

Figure 15: Qualitative comparison of saliency maps calculation methods on the supervised classification VGG16 model (Simonyan & Zisserman, 2014) applied to the ImageNet dataset (Deng et al., 2009).

## F   Self-supervised models image segmentation based on CAMs

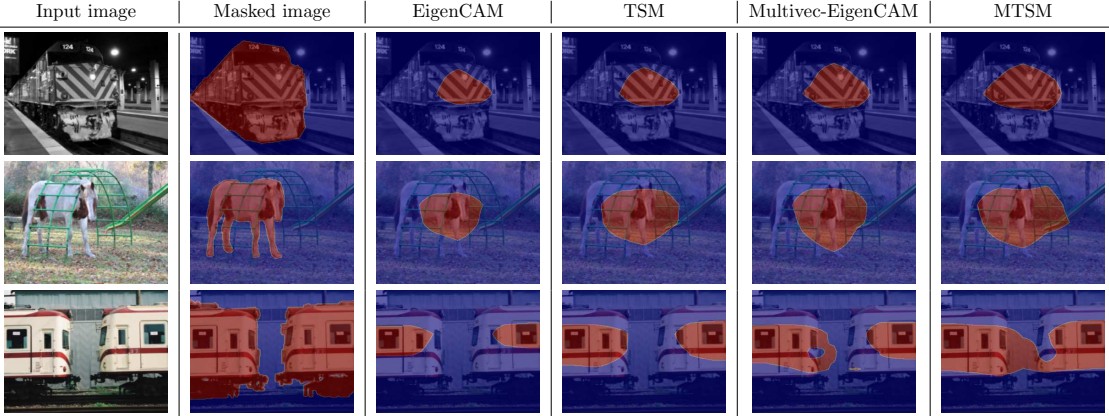

Figure 16: Qualitative comparison of the binarized saliency maps extracted using the EigenCAM method versus the TSM, Multivec-EigenCAM and MTSM methods on the VicRegL Resnet50 model (Bardes et al., 2022), juxtaposed with the segmentation mask from the Pascal VOC dataset (Everingham et al., 2010). The binarization is performed using a 0.5 threshold.

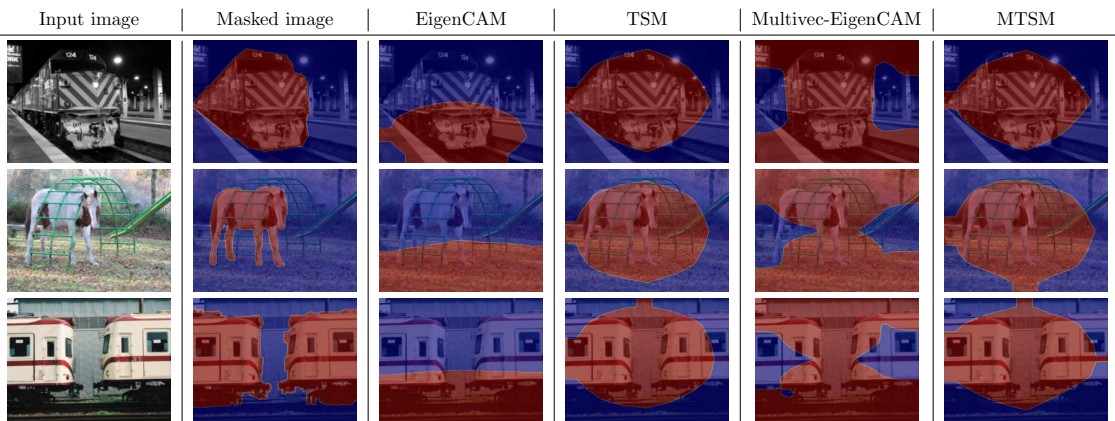

Figure 17: Qualitative comparison of the binarized saliency map extracted using the EigenCAM method versus the TSM, Multivec-EigenCAM and MTSM methods on the VicRegL ConvNext model (Bardes et al., 2022), juxtaposed with the segmentation mask from the Pascal VOC dataset (Everingham et al., 2010). The binarization is performed using a 0.5 threshold.

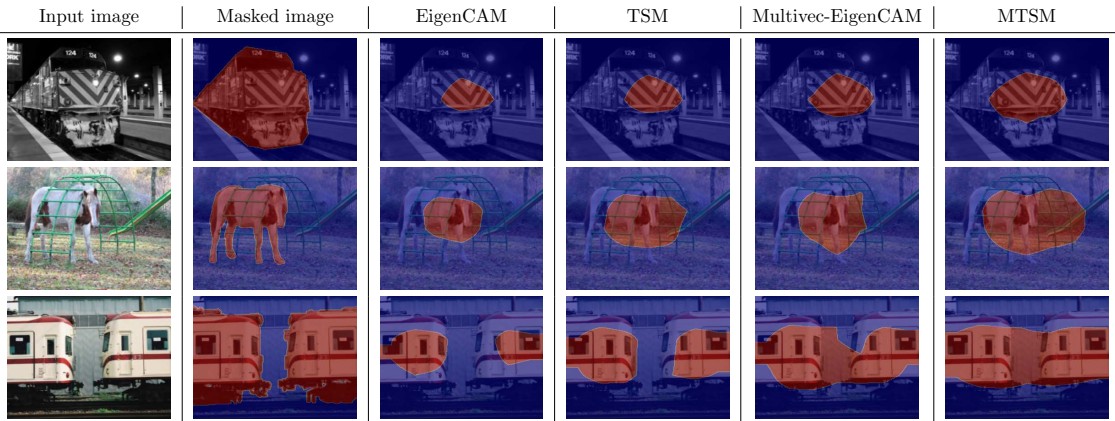

Figure 18: Qualitative comparison of the binarized saliency map extracted using the EigenCAM method versus theTSM, Multivec-EigenCAM and MTSM methods on the Barlow Twins model (Zbontar et al., 2021), juxtaposed with the segmentation mask from the Pascal VOC dataset (Everingham et al., 2010). The binarization is performed using a 0.5 threshold.

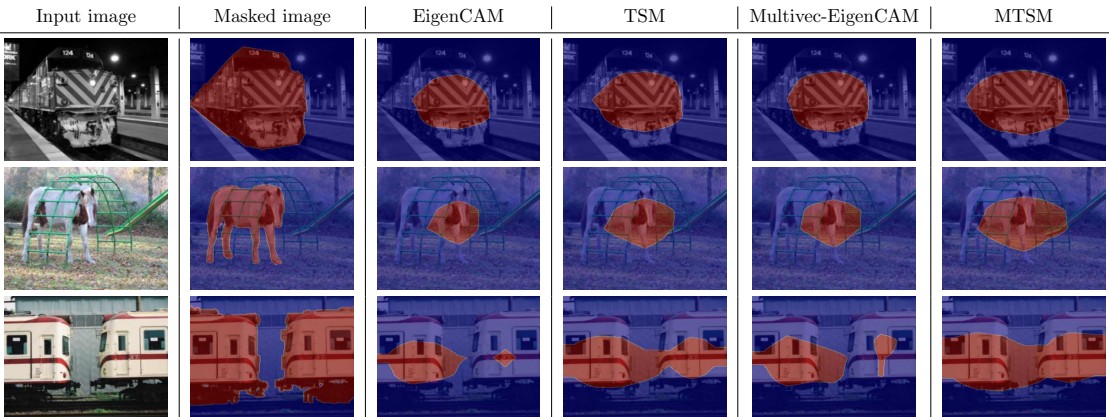

Figure 19: Qualitative comparison of the binarized saliency map extracted using the EigenCAM method versus the TSM, Multivec-EigenCAM and MTSM methods on the Moco V2 model (Chen et al., 2020b), juxtaposed with the segmentation mask from the Pascal VOC dataset (Everingham et al., 2010). The binarization is performed using a 0.5 threshold.

| Input image | Masked image | EigenCAM | TSM | Multivec-EigenCAM | MTSM |
| --- | --- | --- | --- | --- | --- |

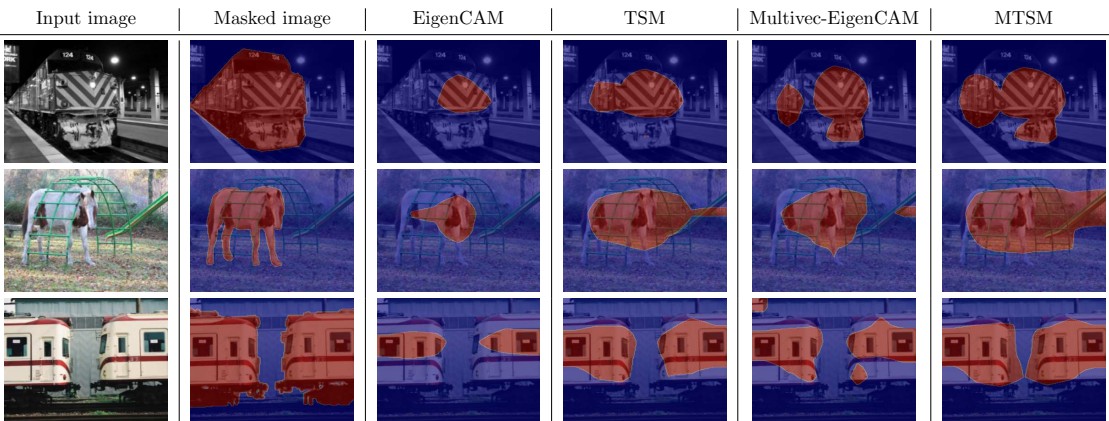

Figure 20: Qualitative comparison of the binarized saliency map extracted using the EigenCAM method versus the TSM, Multivec-EigenCAM and MTSM methods on the SWAV model (Caron et al., 2020), juxtaposed with the segmentation mask from the Pascal VOC dataset (Everingham et al., 2010). The binarization is performed using a 0.5 threshold.

