# OpenReview forum: "CNN Interpretability with Multivector Tucker Saliency Maps for Self-Supervised Models"
_TMLR — Accepted by TMLR_

### Review · Reviewer_94v7 · 2024-11-18

**Summary Of Contributions:**

The paper proposes a method for improving interpretability (note: the authors use 'explainability') of CNNs using a technique from Tensor calculus, i.e., multi-dimensional extensions of matrix decompositions. This extension is especially useful in the context of unsupervised learning (e.g., clustering) as it is class-independent. The authors extend their method, as well as the method EigenCAM from literature into a multivector version, which uses a weighted sum of vectors from the SVD/Tucker decomposition to capture more of the variance and produce more useful saliency maps.

**Audience:**

Yes

**Claims And Evidence:**

Yes

**Requested Changes:**

1. In Table 1, some of the numbers don't agree with the existing literature (see the GradCam++ paper, https://arxiv.org/pdf/1710.11063), table 10. Could you please explain why the results are different?
1. Please strongly consider replacing "explainability" with "interpretability" everywhere (see, for instance, Section 2.1 of Linardatos, P.; Papastefanopoulos, V.; Kotsiantis, S. Explainable AI: A Review of Machine Learning Interpretability Methods. Entropy 2021)
2. I agree with the authors' basic premise that the majority of saliency attribution methods require a reference class and thus are unsuitable for, e.g., clustering algorithms. However, please note that these methods can be used with reference to any class, not necessarily the correct one, and so the labels are not required. I would ask the authors to update their papers to reflect this distinction. (e.g., in the last sentence of page 1 - "Most existing techniques rely heavily on ground truth labels to generate saliency maps".
3. It would be helpful if the authors explained right away that by "tensors" they simply mean 'multidimensional arrays', since Tensors also refer to the actual multilinear map (rather than just its numerical representation)
4. (Minor) - on the bottom of page 4, please justify why "gradient information is inherently noisy" and "such loss depends heavily on the batch size"
5. For Figures 4, 5, and 6, please also provide the raw image, as it is hard (even on an ipad) to see what's under the saliency maps.

**Strengths And Weaknesses:**

## Strengths

The method is clearly laid out and the background section is useful to understand the method. The advantages of the TSM and MTSM methods are clearly explained and the experimental data shows improvements. Further, the authors evaluate their method also in the context of supervised learning, which is helpful to understand the strengths and weaknesses of the proposed methods.

## Weaknesses

The paper has no major weaknesses. As the authors acknowledge, the new TSM and MTSM methods are most effective in the case of unsupervised learning, but this is expected and discussed.

There are, however, some errors that I believe should be addressed; please see the requested changes section below.

---

> ### Author Response · Authors · 2024-12-06
>
> Dear Reviewer,
>
> We thank you for your positive and thorough review of our work. We address the weaknesses you raised hereafter.
>
> Q1: Indeed, our results differ from those presented in Table 10 of the GradCAM++ paper on the GradCAM and GradCAM++. We suspect that the improved performance we observe in our experiments stems from the difference in the used library (Caffe vs. Pytorch). The Pytorch library is more widely used and hence benefits from more tuned training hyperparameters, which results in improved model weights, thus the saliency maps and the quantitative evaluation.
>
> Q2 : We thank you for the reference, we have adapted the edited version of the paper to take into consideration the distinction between explainability and interpretability.
>
> Q3 : Certainly, as you underlined, label-dependent methods can generate saliency maps without requiring the ground-truth label. However, they do require a reference label/class which isn’t always available as is the case for self-supervised models. This can be seen as another distinction between label-dependent and label-independent methods, which we make more explicit in the edited version of our paper and adapt our text to better address the nuances you raised.
>
> Q4 : We agree, and we added this precision to the introduction of the paper.
>
> Q5 : Regarding “Gradient information is inherently noisy”: Gradient information is noisy because gradients indicate the direction of the steepest descent and since the loss function is non-linear, the gradient can change rapidly. Consequently, we used the word “noisy” as in Section 3, Observation 3 in the EigenCAM paper (https://arxiv.org/abs/2008.00299).
>
> Regarding “such loss depends heavily on the batch size”:
> Contrastive loss depends heavily on the batch size and the number of negative samples provided to estimate the loss accurately. This can be observed in SimCLR’s performance improvement when the batch size is higher. Therefore, the information communicated by the contrastive loss is more accurate when the batch size is high, which is directly correlated with the number of negative samples (See figures B.1 and B.2 in SimCLR paper https://arxiv.org/pdf/2002.05709). Moreover, in Moco, a momentum queue is employed to circumvent the large batch size problem and get a stronger signal during training (See paragraph “Dictionary as a queue” under section 3.2 in https://arxiv.org/abs/1911.05722). Furthermore, Swav also relies on a queue for improved clustering (see paragraph “Working with small batches” under section 3.1 in https://arxiv.org/abs/2006.09882). Consequently, most self-supervised models based on contrastive losses require a large batch size (or a queue to simulate a large batch size) to estimate the loss accurately. However, if we do not have a large batch size (in inference for example) the loss estimation will degrade and the gradient of the loss would be even noisier. Consequently, using this gradient in a GradCAM-like method would result in poor-quality saliency maps.
>
> We thank you for raising this point, and we added these explanations to the manuscript for more clarity in Section 3.2.1
>
> Q6 : We agree, and we have modified the manuscript to incorporate the raw images in all the figures.

---

> > ### Comment · Reviewer_94v7 · 2024-12-06
> > **Thank you for the response**
> >
> > Thank you to the authors for your thorough and well-documented response to my and other reviewer's feedback. I have no further requests or corrections; I think it is a very nice paper!

---

### Review · Reviewer_z2n1 · 2024-11-18

**Summary Of Contributions:**

This paper focuses on features map interpretability in convolutional neural networks. Instead of relying of labels, the authors introduced an unsupervised interpretability method that leverage Tucker decomposition of the features map to be able to capture the direction of higher variances which can be used to highlight objects of interest in images. This Tucker Saliency Map (TSM) is presented as an alternative to another unsupervised interpretability method called EigenCAM which rely on SVD. The authors argue that the Tucker decomposition is better than SVD to approximate the first singular vector with the higher variance. The second contribution of the authors is to introduce a multivector version of EigenCAM and TSM. Instead of using a single singular vector, they compute one saliency map by singular vectors and then perform a weighted sum based on the singular values across the saliency maps. The authors test their methods with supervised and self-supervised models on the following metrics: AD (Average Drop), AI (Average Increase), MSE (Mean-Squared Error), mIoU (Mean Intersection over Union). On both supervised and self-supervised models, the authors show that TSM is better than EigenCAM (same for their multivector version). For supervised training, the performances are still the best for supervised methods such as GradCam. The authors claim to have the state of the art for self-supervised explainability.

**Audience:**

Yes

**Broader Impact Concerns:**

Do not have any Broader Impact Concerns.

**Claims And Evidence:**

Yes

**Requested Changes:**

This is a good paper which I think will be of interest to the TMLR's audience. The claim and evidence seems also to be strong enough. I do not have any strong request except maybe some visualizations in which the object of interest is not in the center and what happen when there are multiple objects from a given class. Same question on having visualization on what happen if there is two different classes in the same image (like a dog + cat).

I would also advise against using images of people for visualization in the paper if the authors did not get explicit consent from them.

**Strengths And Weaknesses:**

Strengths:
- This work is very well introduced and the paper is easy to read. I really appreciate that the authors took the time to write a good background section as well as the relevant related work.
- In addition of leveraging the Tucker decomposition instead of SVD, the author developed a multivector version for both of them.
- The experimental section with both use cases in supervised and self-supervised settings is very relevant. Especially since the authors are leveraging different architectures (Resnet/ConvNext/VGG) and different SSL methods (Moco, SWAV, BT, VICReg)
- It's also great that the authors leverage different metrics such as AD, AI, MSE and mIoU to confirm their findings.
- The authors also provide a lot of visualization between different methods and also a good ablation over the threshold value for mIoU.
- The code is in the appendix as well as the experimental details.
- The authors are correctly addressing the limitations of their work by acknowledging that their method has an increased computational cost.

Weaknesses:
- Most of the models used have been pre-trained on ImageNet which is single object classification (and for which in many case, the object of interest will be close to the center). I am wondering wether this method would be able to find multiple objects in an image. Like all the visualizations we can see in the paper have the object of interest that is in the center of the images. What happen if you have a photo of two cats and each one of them is close to the image border?

---

> ### Author Response · Authors · 2024-12-06
>
> Dear Reviewer,
>
> We thank you for your positive and thorough review of our work. We address the weaknesses you raised hereafter.
>
> In Appendices G and H, we added three images of challenging situations and a discussion. The summary is that when we have one class in the image with multiple subjects, all the saliency map methods can identify the objects in the input image. However, when we have two classes, as in the first image with the dog and the cat at the top and bottom of the image, TSM outperforms EigenCAM as it can locate the cat in the image which EigenCAM fails to do. Furthermore, all the multivector methods can identify all the objects in the image. We opted for separate sections for these images for clarity, however, we can merge them with the images in Appendices D and E in the final version of the paper.
>
> Thank you for your advice on avoiding images of people. We have changed the figures and the text in the paper accordingly.

---

### Review · Reviewer_P129 · 2024-11-23

**Summary Of Contributions:**

The paper Introduced Tucker Saliency Map (TSM), which uses Tucker tensor decomposition to better analyze CNN feature maps' structure compared to existing methods. It can generate more accurate singular vectors and values and create higher-quality saliency maps. The improved MTSM achieves competitive performance with label-dependent methods.

**Audience:**

Yes

**Claims And Evidence:**

Yes

**Requested Changes:**

I think the paper is interesting and does not need much adjustment. Except that the computational cost should be discussed. And the clear limitation on architecture should be explicitly stated.

**Strengths And Weaknesses:**

Strengths:

1. The method can create higher-quality saliency maps

2. MTSM is competitive with label-dependent methods on supervised models.



Weaknesses

1. The method seems to be hard to generalize to other architectures like transformers.

2. The computational cost should be discussed in detail like tables or Figures.

---

> ### Author Response · Authors · 2024-12-06
>
> Dear Reviewer,
>
> We thank you for your positive and thorough review of our work. We address the weaknesses you raised hereafter.
>
> We have added quantitative time estimation for each label-independent method on all the models in Table 3, as well as a discussion of the results. Moreover, we have included in the perspective section the non-trivial future work of extending the TSM and MTSM to the transformer architecture in the conclusion of our work.

---

> > ### Comment · Reviewer_P129 · 2025-01-04
> >
> > Thank you for the response. I have no further questions.

---

### Decision · Action_Editor_G35d · 2025-01-14

**Recommendation:** Accept as is

**Comment:**

Altogether, the reviewers are all positive about this work. In their final recommendations, they acknowledge that the work is interesting and provides a clear contribution to the field of CNN interpretability.

**Audience:**

All the reviewers acknowledge that this work would be of interest to the TMLR audience.

**Claims And Evidence:**

All the reviewers acknowledge that the claims are supported by convincing evidence.